# Flow Matching Calibration for Simulation-Based Inference under Model Misspecification

**Pierre-Louis Ruhlmann** [1] **Michael Arbel** [1] **Florence Forbes** [1] **Pedro L. C. Rodrigues** [1]

## Abstract

Simulation-based inference (SBI) is transforming experimental sciences by enabling parameter estimation in complex non-linear models from simulated data. A persistent challenge, however, is model misspecification. In a Bayesian setting, targeting posterior distributions, errors may arise from the simulator, the noise or prior modelling. These model components are only approximations of reality, and severe mismatches can yield biased or overconfident posteriors. We address this issue by introducing Flow Matching Corrected Posterior Estimation (FMCPE), a framework that leverages the flow matching paradigm to refine simulation-trained posterior estimators using a small set of calibration samples. Our approach proceeds in two stages: first, a posterior approximator is trained on abundant simulated data; second, flow matching transports its predictions toward the true posterior supported by calibration observations. We rely on the later to guide the correction, without requiring explicit knowledge of the misspecification form or of which model components are affected. This design enables FMCPE to combine the scalability of SBI with robustness to distributional shift. Across synthetic benchmarks and real-world datasets, we show that our proposal consistently mitigates the effects of misspecification, delivering improved inference accuracy and uncertainty quantification compared to standard SBI baselines, while remaining computationally efficient.

[1]Univ. Grenoble Alpes, Inria, CNRS, Grenoble INP, LJK, 38000 Grenoble, France. Correspondence to: Firstname Lastname <firsname.lastname@inria.fr>.

*Proceedings of the $43^{rd}$ International Conference on Machine Learning*, Seoul, South Korea. PMLR 306, 2026. Copyright 2026 by the author(s).

## 1. Introduction

Many fields of science and engineering describe complex phenomena using stochastic models, capturing inherent sources of randomness, such as measurement noise and probabilistic dynamics. In a Bayesian setting, these models can be decomposed into two components, a likelihood over data $\boldsymbol{x} \in \mathbb{R}^d$ and a prior distribution over some parameters of interest $\boldsymbol{\theta} \in \mathbb{R}^p$. Mechanistic simulators are then often used to produce synthetic data $\boldsymbol{x}$ given parameters $\boldsymbol{\theta}$, but they rarely yield tractable likelihoods, making classical Bayesian inference methods such as Markov chain Monte Carlo (MCMC) (Robert & Casella, 2005) or variational inference (Rezende & Mohamed, 2015) inapplicable. Simulation-Based Inference (SBI) (Deistler et al., 2025; Cranmer et al., 2020) addresses this limitation by performing Bayesian parameter inference directly from simulated datasets, bypassing the need for an explicit likelihood $p_{\boldsymbol{X}|\boldsymbol{\Theta}}$.

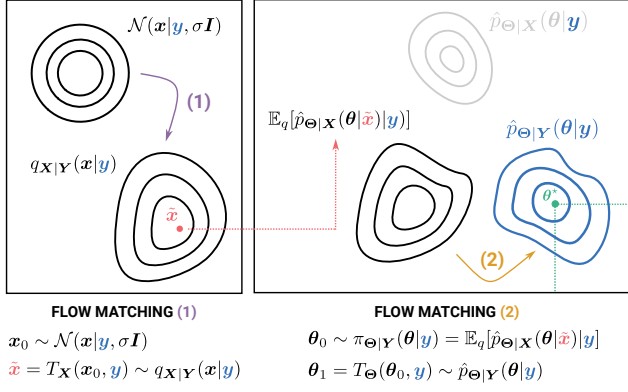

$x_0 \sim \mathcal{N}(\boldsymbol{x}|y, \sigma\boldsymbol{I})$
$\tilde{x} = T_{\boldsymbol{X}}(x_0, y) \sim q_{\boldsymbol{X}|\boldsymbol{Y}}(\boldsymbol{x}|y)$

$\boldsymbol{\theta}_0 \sim \pi_{\boldsymbol{\Theta}|\boldsymbol{Y}}(\boldsymbol{\theta}|y) = \mathbb{E}_q[\hat{p}_{\boldsymbol{\Theta}|\boldsymbol{X}}(\boldsymbol{\theta}|\tilde{x})|y]$
$\boldsymbol{\theta}_1 = T_{\boldsymbol{\Theta}}(\boldsymbol{\theta}_0, y) \sim \hat{p}_{\boldsymbol{\Theta}|\boldsymbol{Y}}(\boldsymbol{\theta}|y)$

*Figure 1.* Overview of FMCPE. The method combines two complementary flow matching steps to correct simulation-based posterior distributions under model misspecification (represented by $\hat{p}_{\boldsymbol{\Theta}|\boldsymbol{X}}(\boldsymbol{\theta}|\boldsymbol{y})$ in grey level sets). (1) Scarce calibration data $(\boldsymbol{\theta}, \boldsymbol{y})$ are used to learn a transport map $T_{\boldsymbol{X}}$ that couples real observations $\boldsymbol{y}$ with surrogate counterparts $\tilde{\boldsymbol{x}}$ lying in the simulator's domain. (2) We then learn $T_{\boldsymbol{\Theta}}$ to transport samples from a $q_{\boldsymbol{X}|\boldsymbol{Y}}$-weighted version of the simulation-based posterior $\hat{p}_{\boldsymbol{\Theta}|\boldsymbol{X}}(\boldsymbol{\theta}|\tilde{\boldsymbol{x}})$ toward the final corrected posterior $\hat{p}_{\boldsymbol{\Theta}|\boldsymbol{Y}}$. Note that both transports are required: $T_{\boldsymbol{X}}$ addresses the mismatches between simulated data and real observations, while $T_{\boldsymbol{\Theta}}$ refines parameter inference to align with the true posterior.

In particular, we consider SBI applications where both $\boldsymbol{\theta}$ values and observations are quantities of physical meaning and interest, but for which it is impractical to model every detail of their generative process. This may generate model mispecification, a notion that needs to be tailored to our SBI setting. The most standard definition of model misspecification in statistics, in both Bayesian (Walker, 2013; Kelly et al., 2024) and frequentist (Tomaselli et al., 2025; Park et al., 2025) contexts, corresponds to the so-called $\mathcal{M}$-open setting of Bernardo & Smith (1994). A model is said to be misspecified ($\mathcal{M}$-open scenario) if there is no $\boldsymbol{\theta}$ value for which the assumed likelihood is able to reproduce the observed data. In this case, there is no such a thing as a *true* parameter $\boldsymbol{\theta}$. The inference target is then a so-called pseudo-true or projection parameter which minimizes a chosen discrepancy to the true data generating process, typically the Kullback-Leibihood divergence in a maximum likeikhood context (Tomaselli et al., 2025; Park et al., 2025; Walker, 2013; Kelly et al., 2024). Unfortunately, this value is in general not practically useful or desirable in SBI contexts where $\boldsymbol{\theta}$ has a real-world meaning. In contrast, in the converse $\mathcal{M}$-closed scenario, there exists a value of $\boldsymbol{\theta}$ for which the likelihood can reproduce the observed data but unfortunately, this property may not be a meaningful guarantee in SBI. As illustrated in Section 2.2 and Appendix A.2 of Wehenkel et al. (2025) with a simple conjugate Gaussian example, the simulator may be well-specified according to the standard $\mathcal{M}$-closed definition but still provides biased estimates of the $\boldsymbol{\theta}$ of interest when applied to real data. See Appendix A for more details. Other definitions of misspecification are more inclusive such as that of Nott et al. (2024) where a model is said to be misspecified if *we are not happy to act as if it is correct*. See also Bürkner et al. (2023) for a more extensive discussion on what is *expected* from a well-specified model. Effective approaches, particularly adapted to ABC settings, to robustify inference include to exhibit summary statistics that mitigate the full data model misspecification (Frazier et al., 2020; Huang et al., 2023; Nott et al., 2024; Kelly et al., 2025). Typically insufficient summary statistics may be compatible with several models but designing such summaries is not the focus of our work.

We thus adapt the notion of misspecification to our SBI context. Considering real or high-fidelity observations $\boldsymbol{y} \in \mathbb{R}^d$, we assume the existence of an unknown or ideal data-generating process, with a likelihood $p_{\boldsymbol{Y}|\boldsymbol{\Theta}}$ and prior $p_{\boldsymbol{\Theta}}$ over parameters of interest. SBI algorithms provide various approaches (Papamakarios & Murray, 2016; Lueckmann et al., 2017; Boelts et al., 2022; Hermans et al., 2020) to obtain approximate samples from the posterior $p_{\boldsymbol{\Theta}|\boldsymbol{Y}} \propto p_{\boldsymbol{\Theta}} \, p_{\boldsymbol{Y}|\boldsymbol{\Theta}}$ with the aid of deep generative models. In this paper, we focus on neural-based approaches that directly approximate the posterior distribution, often called neural posterior estimation (NPE). This corresponds to the

following form of posterior estimates, where the simulator's likelihood $p_{\boldsymbol{X}|\boldsymbol{\Theta}}$ and an assumed prior $q_{\boldsymbol{\Theta}}(\boldsymbol{\theta})$ are used in place of the target data likelihood $p_{\boldsymbol{Y}|\boldsymbol{\Theta}}$ and prior $p_{\boldsymbol{\Theta}}$,

$$\hat{p}_{\boldsymbol{\Theta}|\boldsymbol{X}}(\boldsymbol{\theta}|\boldsymbol{y}) \propto q_{\boldsymbol{\Theta}}(\boldsymbol{\theta}) p_{\boldsymbol{X}|\boldsymbol{\Theta}}(\boldsymbol{y}|\boldsymbol{\theta}) \,. \tag{1}$$

Following Schmitt et al. (2024), we use the word misspecification to refer to the *simulation gap* that may occur when the (simulator) operational generative model assumed during training differs from the (unknown) one inverted at test time. Misspecification may then arise from any of the assumed model components. We will refer to a model as misspecified if any of the model components is undergoing an unacceptably large mismatch. Discrepancies between $p_{\boldsymbol{Y}|\boldsymbol{\Theta}}$ and $p_{\boldsymbol{X}|\boldsymbol{\Theta}}$, can severely degrade parameter inference using the above $\hat{p}_{\boldsymbol{\Theta}|\boldsymbol{X}}$. Likewise, error in prior modelling using $q_{\boldsymbol{\Theta}}$ instead of $p_{\boldsymbol{\Theta}}$ may bias posteriors, particularly in finite-sample settings. Unaccounted noise or contamination can lead to biased or overconfident posterior distributions, ultimately undermining inference reliability (Frazier et al., 2020; Schmitt et al., 2024). Misspecification generally reflects limitations of the simulator or surrogate forward model—whether due to mathematical approximations, simplified dynamics, unmodeled noise, or insufficient computational resources to run more realistic simulations. In particular, modern SBI methods built on deep generative models are especially vulnerable to misspecification. They often perform poorly when faced with out-of-distribution data (Nalisnick et al., 2019) and can fail dramatically across a wide range of real-world problems (Cannon et al., 2022; Schmitt et al., 2024).

A natural way to address imperfections in $\hat{p}_{\boldsymbol{\Theta}|\boldsymbol{X}}$ is to act to reduce its deviation from an estimator built on high-quality data. Following recent literature (Wehenkel et al., 2025; Senouf et al., 2025; Krouglova et al., 2026), we assume access to such accurate representations via a calibration dataset of parameter–observation pairs $\mathcal{D}_{\text{cal}} = \{(\boldsymbol{\theta}_j, \boldsymbol{y}_j)\}_{1 \leq j \leq N_{\text{cal}}}$, obtained either from high-fidelity simulations or from ground-truth controlled experiments. In experimental sciences, experts typically collect such calibrated field data to offset limits in models and sustain the development of accurate digital twins. However, since such data may be scarce due to their potential prohibitive cost, the central challenge is to design strategies that maximally exploit the limited information they provide in order to correct $\hat{p}_{\boldsymbol{\Theta}|\boldsymbol{X}}$.

We propose a calibration method that leverages the flow matching paradigm of Lipman et al. (2023) to refine posterior estimators trained on simulations using only a small set of calibration samples linking parameters with real observations or highly accurate simulations. The principled ability of flows to efficiently model paths between distributions makes them good candidates to design corrective procedures agnostic to the type of misspecifications. Moreover, flows have demonstrated high-scalability with state-of-the-

art performance at large-scale image generation (Esser et al., 2024) and data-efficient performance in SBI (Wildberger et al., 2023). Our approach proceeds in two stages. First, a posterior approximator $\hat{p}_{\Theta|X}$ is trained on abundant simulated data via NPE. Second, a reweighted version of this estimator is used to define a proposal distribution $\pi_{\Theta|Y}$, which serves as the source in a flow matching model that learns a transport map to the true posterior $p_{\Theta|Y}$. Crucially, this transport does not require explicit knowledge of the misspecification form: it aligns the simulation-based posterior with the true posterior supported by real observations. The resulting procedure enables accurate estimation of $p_{\Theta|Y}$ despite the limited availability of calibration data. Figure 1 gives an overview of our method, which we call "Flow Matching for Corrected Posterior Estimation" (FMCPE). Experiments on synthetic and real-world tasks show that FMCPE is more robust to misspecification than standard SBI baselines, while being computationally efficient, amortized and applicable as a post-processing to any SBI model.

The paper is organized as follows. We begin with an overview of SBI under model misspecification and outline how it relates to our contribution. Next, we motivate and detail our methodology, introducing flow matching concepts and notation as needed. Finally, we present numerical experiments and conclude with a discussion of the results.

## 2. Related work

Model misspecification in SBI has been studied both in the framework of Approximate Bayesian Computation (ABC) (Frazier et al., 2020; Bharti et al., 2022; Fujisawa et al., 2021) and with modern neural-based approaches (Kelly et al., 2024; Ward et al., 2022; Huang et al., 2023; Schmitt et al., 2024). These works focus on settings distinct from ours. For instance, Ward et al. (2022) assume a known form of misspecification, while Huang et al. (2023) penalize the NPE loss according to the distributional shift between summary statistics of $x$ and $y$. Other contributions, such as Schmitt et al. (2024), restrict attention to detecting misspecification at inference time by imposing a Gaussian prior on the space of summary statistics.

An alternative perspective is to interpret misspecification as a noisy channel between simulated and real-world observations. In this view (Wehenkel et al., 2025; Ward et al., 2022), once simulated data $x$ are known, the real observation $y$ does not carry additional information about the parameters $\theta$. Under this conditional independence hypothesis

$$y \perp \theta \mid x,  \qquad (2)$$

the true posterior distribution can be expressed as

$$p_{\Theta|Y}(\theta|y) = \int p_{\Theta|X}(\theta|x) p_{X|Y}(x|y) \, \mathrm{d}x, \qquad (3)$$

which states that the true posterior distribution is a mixture of surrogate posteriors, weighted by the probability of each simulated outcome $x$ given $y$. Wehenkel et al. (2025) exploit this idea by assuming access to a ground truth calibration dataset Their method, named RoPE, approximates $p_{\Theta|Y}$ by replacing $p_{X|Y}$ in Equation (3) with a coupling $q_{X|Y}$ estimated via optimal transport (Peyré & Cuturi, 2019), which links real observations $y$ and simulated data $x$. A key limitation is that RoPE requires access to a full batch of test samples at inference time, preventing its use in online or sequential prediction scenarios. This issue has been addressed by Senouf et al. (2025). However, both approaches rely on the conditional independence assumption (2), which does not always hold. For example, if model misspecification stems from the fact that the simulator neglects some parameters, for $\theta = (\theta_1, \theta_2)$, $p_{X|\Theta}(x|\theta) = f(\theta_1)$ while $p_{Y|\Theta}(y|\theta) = g(\theta_1, \theta_2)$, then $y$ will still depend on the full parameter vector, violating conditional independence.

In contrast, the method MFNPE proposed by Krouglova et al. (2026) does not rely on conditional independence nor require access to the full test set. Their approach first trains a posterior approximator $p_{\mathrm{LF}}(\theta|x)$ on low-fidelity simulations using standard NPE, then refines it with high-fidelity samples (i.e. coming from high-fidelity simulators or real data). Similarly, our method does not assume (2) or any other conditional independence and does not make any specific assumptions on the misspecification form. Our approach follows the same spirit of leveraging low-fidelity approximations obtained from a simulator, but uses them differently: the low-fidelity estimator plays a distinct role, and the subsequent corrections it undergoes are of a different nature, as detailed in the next section.

## 3. Flow Matching Corrected Posterior Estimation

We consider the general problem of sampling from a posterior distribution $p_{\Theta|Y}$ resulting from an *unknown* likelihood $p_{Y|\Theta}$ over observation $y \in \mathbb{R}^d$, and prior $p_\Theta$ over a parameter vector $\theta \in \mathbb{R}^p$. If the model were known, the posterior would be available using the Bayes rule as $p_{\Theta|Y}(\theta|y) \propto p_{Y|\Theta}(y|\theta)p_\Theta(\theta)$. Instead, we assume only access to a potentially approximate prior $q_\Theta$ on $\theta$ and to an *imperfect* stochastic simulator $S : (\theta, \epsilon) \mapsto x$, generating low cost simulations $x \in \mathbb{R}^d$, with $\epsilon$ a random noise accounting for randomness in the generative process. The simulator implicitly defines a generative model $p_{X|\Theta}$ whose posterior distribution can be approximated by an easy-to-sample model $\hat{p}_{\Theta|X}$, using techniques from the SBI literature such as NPE. Unfortunately, since the simulator or prior can be inaccurate, the SBI model $\hat{p}_{\Theta|X}$ is likely not to provide an accurate approximation to the true posterior $p_{\Theta|Y}$, in general, no matter how accurately it approximates the simu-

lator's posterior $p_{\Theta|X}$. However, one can reasonably expect such learned model to be informative about the true posterior. We propose to leverage the simulator $S$ and posterior model $\hat{p}_{\Theta|X}$ in addition to a small set of calibration pairs of high-quality data $\mathcal{D}_{\text{cal}} = \{(\boldsymbol{\theta}_j, \boldsymbol{y}_j)\}_{1 \leq j \leq N_{\text{cal}}}$ from the joint distribution $p_{\Theta, Y}$ to provide an accurate and efficient model $\hat{p}_{\Theta|Y}$ for the true posterior $p_{\Theta|Y}$. Consequently, we assume that $N_{\text{cal}}$ is not large enough to provide an accurate posterior estimate from this dataset alone.

We propose to use the flow matching paradigm to learn a dynamic transport map, a vector field, from a carefully designed source distribution $\pi_{\Theta|Y}$ towards the target posterior $p_{\Theta|Y}$ using the calibration dataset $\mathcal{D}_{\text{cal}}$. Increasing the proximity of the source and target distributions reduces the complexity of the flow and makes it easier to learn from a small number of samples. Intuitively, smaller distributional gaps are likely to require fewer steps and improve sample complexity (Cui et al., 2024; Lin et al., 2025; Kong et al., 2025; Wang et al., 2025). Following this principle, we employ the simulator $S$ and the posterior approximation $\hat{p}_{\Theta|X}$, to design and train an informative source distribution that facilitates the learning of the vector field from few calibration samples. The resulting algorithm (Algorithm 1 in Section 3.3) combines simultaneous learning of both the source distribution and the vector field, ensuring the source is continually updated as the vector field becomes more accurate. Next, we detail the steps for learning the vector field given an informed source distribution, constructing the source itself and training them jointly.

### 3.1. Data-efficient posterior flow matching

The calibration set $\mathcal{D}_{\text{cal}}$ is used to learn a transport map $T_{\Theta} : \mathbb{R}^p \times \mathbb{R}^d \to \mathbb{R}^p$ from an easy-to-sample base (or source) distribution $\pi_{\Theta}$ towards the posterior $p_{\Theta|Y}$:

$$\boldsymbol{\theta}_0 \sim \pi_{\Theta}(\boldsymbol{\theta}) \Rightarrow \boldsymbol{\theta}_1 = T_{\Theta}(\boldsymbol{\theta}_0, \boldsymbol{y}) \sim p_{\Theta|Y}(\boldsymbol{\theta}|\boldsymbol{y}) . \quad (4)$$

To learn $T_{\Theta}$, we use the flow matching framework of Lipman et al. (2023), an approach that has proven effective for modeling complex target distributions, such as images (Esser et al., 2024), and has recently been applied successfully in simulation-based inference (Wildberger et al., 2023). Flow matching consists of learning a time-dependent vector field $u_{\Theta}$ capable of transporting a sample $\boldsymbol{\theta}$ from $\pi_{\Theta}$ along a trajectory $(\boldsymbol{\theta}_t)_{t \in [0,1]}$ starting from $\boldsymbol{\theta}_0 = \boldsymbol{\theta}$, so that $\boldsymbol{\theta}_1$ is distributed according to the target $p_{\Theta|Y}$. The trajectory is obtained as a solution of the following ODE:

$$\frac{\mathrm{d}\boldsymbol{\theta}_t}{\mathrm{d}t} = u_{\Theta}(t, \boldsymbol{\theta}_t, \boldsymbol{y}), \quad \forall t \in [0,1], \quad \text{with } \boldsymbol{\theta}_0 = \boldsymbol{\theta}.$$

The time-dependent flow $\psi_{\Theta}$ associated to the above ODE is simply given by samples along the path starting from the initial condition $\boldsymbol{\theta}$, i.e. $\psi_{\Theta}(t, \boldsymbol{\theta}, \boldsymbol{y}) := \boldsymbol{\theta}_t$. Such a

flow allows defining the transport map from $\pi_{\Theta}$ to $p_{\Theta|Y}$ as $T_{\Theta}(\boldsymbol{\theta}, \boldsymbol{y}) := \psi_{\Theta}(1, \boldsymbol{\theta}, \boldsymbol{y})$. The vector field is approximated with a deep neural network $\hat{u}_{\Theta}$ trained on a *guided* version of the conditional flow-matching loss from Lipman et al. (2023) with a linear interpolation strategy:

$$\mathcal{L}_{\Theta}(\hat{u}_{\Theta}) = \mathbb{E}\left[ \int_0^1 \|\hat{u}_{\Theta}(t, \boldsymbol{\theta}_t, \boldsymbol{y}) - (\boldsymbol{\theta}_1 - \boldsymbol{\theta}_0)\|^2 \, \mathrm{d}t \right],$$
$$\boldsymbol{\theta}_t := (1 - t)\boldsymbol{\theta}_0 + t\boldsymbol{\theta}_1 \, \forall t \in [0, 1] ,$$
$$(5)$$

where the expectation is taken over $(\boldsymbol{\theta}_1, \boldsymbol{y}, \boldsymbol{\theta}_0) \sim p_{\Theta, Y}(\boldsymbol{\theta}_1, \boldsymbol{y})\pi_{\Theta}(\boldsymbol{\theta}_0)$. The calibration dataset $\mathcal{D}_{\text{cal}}$ can be used to provide joint samples $(\boldsymbol{\theta}_1, \boldsymbol{y})$ from $p_{\Theta, Y}(\boldsymbol{\theta}_1, \boldsymbol{y})$, thus providing an empirical version of the above objective.

In most flow matching instances, the base distribution $\pi_{\Theta}$ is set to a simple standardized Gaussian distribution, i.e. $\pi_{\Theta}(\boldsymbol{\theta}) = \mathcal{N}(\boldsymbol{\theta} \mid \boldsymbol{0}, \boldsymbol{I}_p)$. However, for settings such as the one we consider here, where $\mathcal{D}_{\text{cal}}$ is small, this choice may result in very poor posterior approximators with high variance. Instead, we use the simulator to construct a more informative data-driven source distribution $\pi_{\Theta} = \pi_{\Theta|Y}$ that acts as a possibly low-quality surrogate of the true $p_{\Theta|Y}$. Training itself remains consistent with standard guided flow matching; the learned flow then serves to refine this source distribution using only the limited calibration data.

### 3.2. Simulation-informed source distribution

The source distribution $\pi_{\Theta|Y}$ plays a central role in our framework. Ideally, it should be very close to the true posterior distribution, so that the flow $\psi_{\Theta}$ induces minimal corrections to $\boldsymbol{\theta}_0 \sim \pi_{\Theta|Y}$. Making use of the availability of the simulator $S$, a first natural possibility is to set $\pi_{\Theta|Y}(\boldsymbol{\theta}|\boldsymbol{y}) = \hat{p}_{\Theta|X}(\boldsymbol{\theta}|\boldsymbol{y})$ where $\hat{p}_{\Theta|X}$ is a posterior approximation obtained via SBI. This provides a reasonable approximation of the true $p_{\Theta|Y}$ in the absence of misspecification but is likely to be a poor one in less favorable settings. A natural alternative is to plug into $\hat{p}_{\Theta|X}$ a conditioning sample from a distribution $q_{X|Y}(\boldsymbol{x}|\boldsymbol{y})$ with the same support as $p_X(\boldsymbol{x})$ and informative of $\boldsymbol{y}$. This corresponds to considering a source distribution of the form

$$\pi_{\Theta|Y}(\boldsymbol{\theta}|\boldsymbol{y}) = \int \hat{p}_{\Theta|X}(\boldsymbol{\theta}|\boldsymbol{x}) \, q_{X|Y}(\boldsymbol{x}|\boldsymbol{y}) \, d\boldsymbol{x}. \quad (6)$$

The most straightforward choice for $q_{X|Y}$ is $q_{X|Y}(\boldsymbol{x}|\boldsymbol{y}) := p_{X|Y}(\boldsymbol{x}|\boldsymbol{y})$, which would require, in addition to $\mathcal{D}_{\text{cal}}$, a large number of ground truth samples $(\boldsymbol{y}, \boldsymbol{x})$ for training an approximator. Thus, we consider instead,

$$q_{X|Y}(\boldsymbol{x}|\boldsymbol{y}) := \int p_{X|\Theta}(\boldsymbol{x}|\boldsymbol{\theta}) \, p_{\Theta|Y}(\boldsymbol{\theta}|\boldsymbol{y}) \, d\boldsymbol{\theta}, \quad (7)$$

that can be seen as a mixture and can be approximated using simulator $S$ and joint calibration pairs $(\boldsymbol{\theta}_j, \boldsymbol{y}_j) \sim$

$p_{\boldsymbol{\Theta},\boldsymbol{Y}}(\boldsymbol{\theta},\boldsymbol{y}) = p_{\boldsymbol{\Theta}|\boldsymbol{Y}}(\boldsymbol{\theta}|\boldsymbol{y})p_{\boldsymbol{Y}}(\boldsymbol{y})$. Indeed, note that for each $\boldsymbol{y}_j \in \mathcal{D}_{\mathrm{cal}}$, its associated $\boldsymbol{\theta}_j$ is a sample from $p_{\boldsymbol{\Theta}|\boldsymbol{Y}}(\boldsymbol{\theta}|\boldsymbol{y}_j)$ that can be plugged in the simulator to generate $\boldsymbol{x}_j \sim p_{\boldsymbol{X}|\boldsymbol{\Theta}}(\boldsymbol{x}|\boldsymbol{\theta}_j)$, leading to $\boldsymbol{x}_j \sim q_{\boldsymbol{X}|\boldsymbol{Y}}(\boldsymbol{x}|\boldsymbol{y}_j)$. This is typically used in lines 1 and 2 of Algorithm 2. Note that $q_{\boldsymbol{X}|\boldsymbol{Y}} = p_{\boldsymbol{X}|\boldsymbol{Y}}$ if $\boldsymbol{x} \perp \boldsymbol{y} \mid \boldsymbol{\theta}$, which may not be true if $\boldsymbol{y}$ depends on the nuisance parameter $\epsilon$.

A sample $\boldsymbol{\theta}$ from the source distribution (6), can be obtained by first drawing a sample $\tilde{\boldsymbol{x}}$ from $q_{\boldsymbol{X}|\boldsymbol{Y}}(\tilde{\boldsymbol{x}}|\boldsymbol{y})$ in (7), then setting $\boldsymbol{\theta}$ to a sample from $\hat{p}_{\boldsymbol{\Theta}|\boldsymbol{X}}(\boldsymbol{\theta}|\tilde{\boldsymbol{x}})$. As we can already easily sample from $\hat{p}_{\boldsymbol{\Theta}|\boldsymbol{X}}$, it only remains to provide a way to sample from $q_{\boldsymbol{X}|\boldsymbol{Y}}(\boldsymbol{x}|\boldsymbol{y})$ for any $\boldsymbol{y}$. To this end, we use the flow matching framework to construct a conditional transport map $T_{\boldsymbol{X}} : \mathbb{R}^d \times \mathbb{R}^d \to \mathbb{R}^d$, from a Gaussian distribution centered around $\boldsymbol{y}$ with isotropic variance $\sigma^2$, towards $q_{\boldsymbol{X}|\boldsymbol{Y}}(\boldsymbol{x}|\boldsymbol{y})$ i.e.:

$$\boldsymbol{x}_0 \sim \mathcal{N}(\boldsymbol{x}|\boldsymbol{y},\sigma^2\boldsymbol{I}_d) \to \tilde{\boldsymbol{x}} = T_{\boldsymbol{X}}(\boldsymbol{x}_0,\boldsymbol{y}) \sim q_{\boldsymbol{X}|\boldsymbol{Y}}(\boldsymbol{x}|\boldsymbol{y}). \quad (8)$$

More specifically, we define $T_{\boldsymbol{X}}(\boldsymbol{x}_0,\boldsymbol{y}) = \psi_{\boldsymbol{X}}(1,\boldsymbol{x}_0,\boldsymbol{y})$, where $\psi_{\boldsymbol{X}}$ is the flow associated to the velocity field neural approximator $\hat{u}_{\boldsymbol{X}}$ learned by minimizing the following loss:

$$\mathcal{L}_{\boldsymbol{X}}(\hat{u}_{\boldsymbol{X}}) = \mathbb{E}\left[\int_0^1 \|\hat{u}_{\boldsymbol{X}}(t,\boldsymbol{x}_t,\boldsymbol{y}) - (\boldsymbol{x}_1 - \boldsymbol{x}_0)\|^2 \,\mathrm{d}t\right]$$
$$\boldsymbol{x}_t := (1-t)\boldsymbol{x}_0 + t\boldsymbol{x}_1 \;\forall t \in [0,1] \quad (9)$$

with $(\boldsymbol{y},\boldsymbol{x}_1,\boldsymbol{x}_0) \sim q_{\boldsymbol{Y},\boldsymbol{X}}(\boldsymbol{y},\boldsymbol{x}_1)\mathcal{N}(\boldsymbol{x}_0|\boldsymbol{y},\sigma^2\boldsymbol{I}_d)$. The dataset $\mathcal{D}_{\mathrm{cal}}$ and simulator $S$ can be used to provide joint samples $(\boldsymbol{y},\boldsymbol{x}_1)$ from $q_{\boldsymbol{Y},\boldsymbol{X}}(\boldsymbol{y},\boldsymbol{x}_1)$ as discussed later in Section 3.3. We took inspiration from Albergo et al. (2024) to define a source distribution that induces a coupling between the base and target distributions through the conditioning variable, which greatly helps the training process under limited data.

When the conditional independence hypothesis (2) is valid and choosing $q_{\boldsymbol{X}|\boldsymbol{Y}}(\boldsymbol{x}|\boldsymbol{y}) = p_{\boldsymbol{X}|\boldsymbol{Y}}(\boldsymbol{x}|\boldsymbol{y})$, we can use Equation (3) to see that $p_{\boldsymbol{X}|\boldsymbol{Y}}(\boldsymbol{x}|\boldsymbol{y})$ is indeed the optimal choice for transporting data from $\boldsymbol{y}$-space to $\boldsymbol{x}$-space, since

$$p_{\boldsymbol{\Theta}|\boldsymbol{Y}}(\boldsymbol{\theta}|\boldsymbol{y}) = \mathbb{E}_{\boldsymbol{X}|\boldsymbol{Y}}\left[p_{\boldsymbol{\Theta}|\boldsymbol{X}}(\boldsymbol{\theta}|\boldsymbol{x})|\boldsymbol{y}\right],$$
$$\tilde{\boldsymbol{x}} \sim p_{\boldsymbol{X}|\boldsymbol{Y}}(\boldsymbol{x}|\boldsymbol{y}) \text{ and } \boldsymbol{\theta} \sim p_{\boldsymbol{\Theta}|\boldsymbol{X}}(\boldsymbol{\theta}|\tilde{\boldsymbol{x}}) \Longrightarrow \boldsymbol{\theta} \sim p_{\boldsymbol{\Theta}|\boldsymbol{Y}}(\boldsymbol{\theta}|\boldsymbol{y}).$$

In that case, learning only $\hat{u}_{\boldsymbol{X}}$ could be sufficient, since the proposal $\pi_{\boldsymbol{\Theta}|\boldsymbol{Y}}$ is already a good approximation to the true posterior. However, if the conditional independence is no longer valid, if $\hat{p}_{\boldsymbol{\Theta}|\boldsymbol{X}}$ is poorly trained, or $\hat{u}_{\boldsymbol{X}}$ is not optimal, the proposal would not have the flexibility to compensate for errors from each of its different parts. This is no longer an issue when using $\hat{u}_{\boldsymbol{X}}$ to define a source distribution for learning the vector field $\hat{u}_{\boldsymbol{\Theta}}$, as proposed in this work, since it does not rely on the validity of Equation (2).

## 3.3. Joint training of posterior and source distributions by flow matching

We now face two optimization tasks: minimizing the loss in (9) to train the simulation-space vector field $\hat{u}_{\boldsymbol{X}}$ and minimizing (5) to train the parameter-space field $\hat{u}_{\boldsymbol{\Theta}}$. Instead of solving them separately, we propose to optimize the following *joint objective* for improved practical performance:

$$\mathcal{L}_{\boldsymbol{X},\boldsymbol{\Theta}}(\hat{u}_{\boldsymbol{X}},\hat{u}_{\boldsymbol{\Theta}}) = \mathbb{E}\left[\int_0^1 \|\hat{u}_{\boldsymbol{\Theta}}(t,\boldsymbol{\theta}_t,\boldsymbol{y}) - (\boldsymbol{\theta}_1 - \boldsymbol{\theta}_0)\|^2 \mathrm{d}t \right.$$
$$\left. + \int_0^1 \|\hat{u}_{\boldsymbol{X}}(t,\boldsymbol{x}_t,\boldsymbol{y}) - (\boldsymbol{x}_1 - \boldsymbol{x}_0)\|^2 \mathrm{d}t\right] \quad (10)$$

with $(\boldsymbol{\theta}_1,\boldsymbol{y}) \sim p_{\boldsymbol{\Theta},\boldsymbol{Y}}$,

$(\boldsymbol{x}_0,\boldsymbol{x}_1,\boldsymbol{\theta}_0) \sim \mathcal{N}(\boldsymbol{x}_0;\boldsymbol{y},\sigma^2\boldsymbol{I}_d)q_{\boldsymbol{X}|\boldsymbol{Y}}(\boldsymbol{x}_1 \mid \boldsymbol{y})\pi_{\boldsymbol{\Theta}|\boldsymbol{Y}}(\boldsymbol{\theta}_0 \mid \boldsymbol{y})$

and $\boldsymbol{\theta}_t$ and $\boldsymbol{x}_t$ are convex combinations as in Equations (5) and (9). In practice, at a given iteration $k$ of the optimization procedure, $(\boldsymbol{\theta}_1,\boldsymbol{y})$ are sampled uniformly from the calibration dataset, where $\boldsymbol{\theta}_1$ serves as target sample to the vector field $\hat{u}_{\boldsymbol{\Theta}}$. We then generate $\boldsymbol{x}_1$ given $\boldsymbol{y}$ using (7), that is $\boldsymbol{x}_1 \sim p_{\boldsymbol{X}|\boldsymbol{\Theta}}(\boldsymbol{x}|\boldsymbol{\theta}_1)$, or equivalently by setting $\boldsymbol{x}_1 = S(\boldsymbol{\theta}_1,\epsilon)$ as explained in Section 3.2. This $\boldsymbol{x}_1$ serves as target for $\hat{u}_{\boldsymbol{X}}$, while $\boldsymbol{x}_0$ given $\boldsymbol{y}$ is obtained from a Gaussian distribution centered at $\boldsymbol{y}$. For the source distribution (6), $\boldsymbol{\theta}_0$ is sampled from $\hat{p}_{\boldsymbol{\Theta}|\boldsymbol{X}}(\boldsymbol{\theta}|\tilde{\boldsymbol{x}})$ where in contrast to $\boldsymbol{x}_1$, $\tilde{\boldsymbol{x}}$ is not sampled from the simulator $S$ but is generated using the ODE induced by the current estimated vector field $\hat{u}_{\boldsymbol{X}}^{(k)}$ and starting from $\boldsymbol{x}_0$, $\tilde{\boldsymbol{x}} = \psi_{\boldsymbol{X}}^{(k)}(1,\boldsymbol{x}_0,\boldsymbol{y})$. This also makes $\boldsymbol{\theta}_0$ independent of $\boldsymbol{\theta}_1$ given $\boldsymbol{y}$. It follows that compared to standard flow matching, $\boldsymbol{\theta}_0$ is sampled from a source $\pi_{\boldsymbol{\Theta}|\boldsymbol{Y}}$ which is evolving with $\hat{u}_{\boldsymbol{X}}^{(k)}$ during training. Intuitively, by coupling the estimations of $\hat{u}_{\boldsymbol{X}}$ and $\hat{u}_{\boldsymbol{\Theta}}$, this joint formulation forces $\hat{u}_{\boldsymbol{\Theta}}$ to be robust to noisy or inaccurate samples from the source distribution: during early training stages, $\pi_{\boldsymbol{\Theta}|\boldsymbol{Y}}$ may yield poor candidates $\boldsymbol{\theta}_0$, yet $\hat{u}_{\boldsymbol{\Theta}}$ must learn to accommodate them. This robustness is desirable at test time, where the source distribution may also generate imperfect samples for previously unseen observations $\boldsymbol{y}$.

The joint learning of vector fields $\hat{u}_{\boldsymbol{X}}$ and $\hat{u}_{\boldsymbol{\Theta}}$ is summarized in Algorithm 1, which specifies how to optimize the objective $\mathcal{L}_{\boldsymbol{X},\boldsymbol{\Theta}}(\hat{u}_{\boldsymbol{X}},\hat{u}_{\boldsymbol{\Theta}})$ using the sampling procedure described above and summarized in Algorithm 2 referred to as function SAMPLETRAININGTUPLE. In practice, variable $t$ is sampled i.i.d. for each of the terms $\boldsymbol{X}$ and $\boldsymbol{\Theta}$ from $\mathcal{L}_{\boldsymbol{X},\boldsymbol{\Theta}}$ (lines 5 and 7) to reduce bias. Also, because $\hat{u}_{\boldsymbol{X}}$ affects the distribution of $\boldsymbol{\theta}_0$ (via $\tilde{\boldsymbol{x}}$), the effective source distribution for $\hat{u}_{\boldsymbol{\Theta}}$ is non-stationary; to mitigate instability we use small learning rates and gradient clipping to reduce training instability in the early stages. Another crucial aspect in the above sampling procedure is to prevent gradients from prop-

agating through the intermediate sample $\tilde{x}$ when optimizing the vector field $\hat{u}_{\boldsymbol{X}}$ which would bias its training.

---

**Algorithm 1** Joint flow training for FMCPE

---

**Require:** SAMPLETRAININGTUPLE; trainable flows $\hat{u}_{\boldsymbol{X}}$, $\hat{u}_{\boldsymbol{\Theta}}$; minibatch size $B$
1: **repeat**
2:  $\quad \mathcal{L} \leftarrow 0$
3:  $\quad$ **for** $i = 1$ to $B$ **do**
4:  $\quad\quad (\boldsymbol{y}, \boldsymbol{\theta}_1, \boldsymbol{\theta}_0, \boldsymbol{x}_1, \boldsymbol{x}_0) \leftarrow$ SAMPLETRAININGTUPLE$(\hat{u}_{\boldsymbol{X}})$
   $\quad\quad$ {see Algorithm 2}
5:  $\quad\quad$ Draw $t \sim \mathcal{U}[0, 1]$
6:  $\quad\quad \boldsymbol{x}_t \leftarrow (1 - t)\boldsymbol{x}_0 + t\,\boldsymbol{x}_1$
7:  $\quad\quad$ Draw *another independent* $\tau \sim \mathcal{U}[0, 1]$
8:  $\quad\quad \boldsymbol{\theta}_\tau \leftarrow (1 - \tau)\boldsymbol{\theta}_0 + \tau\,\boldsymbol{\theta}_1$
9:  $\quad\quad \ell_{\boldsymbol{X}} \leftarrow \|\hat{u}_{\boldsymbol{X}}(t, \boldsymbol{x}_t, \boldsymbol{y}) - (\boldsymbol{x}_1 - \boldsymbol{x}_0)\|^2$
10: $\quad\quad \ell_{\boldsymbol{\Theta}} \leftarrow \|\hat{u}_{\boldsymbol{\Theta}}(\tau, \boldsymbol{\theta}_\tau, \boldsymbol{y}) - (\boldsymbol{\theta}_1 - \boldsymbol{\theta}_0)\|^2$
11: $\quad\quad \mathcal{L} \leftarrow \mathcal{L} + \ell_{\boldsymbol{X}} + \ell_{\boldsymbol{\Theta}}$
12: $\quad$ **end for**
13: $\quad \mathcal{L} \leftarrow \mathcal{L}/B$
14: $\quad$ Update parameters of $\hat{u}_{\boldsymbol{X}}$ and $\hat{u}_{\boldsymbol{\Theta}}$
15: **until** convergence

---

**Algorithm 2** SAMPLETRAININGTUPLE$(\hat{u}_{\boldsymbol{X}})$

---

**Require:** Calibration set $\mathcal{D}_{\text{cal}}$; simulator $S$; pretrained $\hat{p}_{\boldsymbol{\Theta}|\boldsymbol{X}}(\boldsymbol{\theta} \mid \tilde{x})$ (frozen); velocity field $\hat{u}_{\boldsymbol{X}}$
1: Sample $(\boldsymbol{\theta}_1, \boldsymbol{y}) \sim \mathcal{D}_{\text{cal}}$ {calibration sample}
2: Sample $\boldsymbol{x}_1$ using simulator $S$ evaluated at $\boldsymbol{\theta}_1$ {see Equation (7)}
3: Draw base sample $\boldsymbol{x}_0 \sim \mathcal{N}(\boldsymbol{x}_0, \boldsymbol{y}, \sigma^2 I)$ {base distribution}
4: Solve ODE $\frac{d\bar{\boldsymbol{x}}_t}{dt} = \hat{u}_{\boldsymbol{X}}(t, \bar{\boldsymbol{x}}_t, \boldsymbol{y})$ with $\bar{\boldsymbol{x}}_0 = \boldsymbol{x}_0$
5: $\tilde{x} \leftarrow T_{\boldsymbol{X}}(\boldsymbol{x}_0; \boldsymbol{y}) := \bar{\boldsymbol{x}}_1$ {current flow map}
6: $\tilde{x} \leftarrow$ StopGradient$(\tilde{x})$
7: Sample $\boldsymbol{\theta}_0 \sim \hat{p}_{\boldsymbol{\Theta}|\boldsymbol{X}}(\boldsymbol{\theta} \mid \tilde{x})$
8: **return** $(\boldsymbol{y}, \boldsymbol{\theta}_1, \boldsymbol{\theta}_0, \boldsymbol{x}_1, \boldsymbol{x}_0)$

---

## 4. Experiments

We benchmark our method against three baselines, NPE, MFNPE, and RoPE. The comparison is carried out on four tasks described in Section 4.1, using various evaluation metrics specified in Section 4.2. We follow the implementation from (Wehenkel et al., 2025) for NPE and train it using only the limited calibration data $\mathcal{D}_{\text{cal}}$. Note that since the simulator is not used, we expect this baseline NPE to fail as the complexity of the generating process increases. For each experiment, all evaluations are performed on a test set $\mathcal{D}_{\text{test}} = \{(\boldsymbol{\theta}_j, \boldsymbol{y}_j)\}_{1 \leq j \leq N_{\text{test}}}$ with $N_{\text{test}} = 2000$ unless otherwise specified, and each metric is reported for different sizes $N_{\text{cal}}$ of the calibration set. Calibration sets are constructed in an expanding manner, gradually adding new samples to an initial set, to limit the sources of variability in the comparison; see Appendix B for details.

All experiments described below are implemented in Python using pytorch (Paszke et al., 2019) and mlxp (Arbel & Zouaoui, 2024). We also use nflows (Durkan et al.,

2020) and the dingo (Dax et al., 2021) packages for the implementation of normalizing flows. Our code is available at https://github.com/pruhlmann/FMCPE.

### 4.1. Tasks

We first evaluate the methods ability to handle simulator misspecification. Our experiments consist of two synthetic and two real-world tasks. Setup and implementation details can be found in Appendix B.

▶ Gaussian: A multivariate Gaussian model is considered, with $\boldsymbol{\theta} \in \mathbb{R}^3$ and $\boldsymbol{\theta} \sim \mathcal{N}(\mu_\theta, \Sigma_\theta)$. Both $\boldsymbol{x} \in \mathbb{R}^{10}$ and $\boldsymbol{y} \in \mathbb{R}^{10}$ follow multivariate Gaussian distributions centered on different linear combinations of $\boldsymbol{\theta}$.

▶ Pendulum: The damped pendulum (Takeishi & Kalousis, 2021) models the oscillations of a mass around a fixed attachment point. Parameters $\boldsymbol{\theta} = [A, \omega_0]$ are the oscillation amplitude $A$ and the natural frequency $\omega_0$. Simulations are generated from a simplified model that omits friction forces, thus creating a systematic misspecification relative to the real dynamics. Both observations and simulations are real-valued time-series of length 200.

▶ Wind Tunnel: This task from Gamella et al. (2025) consists of measuring the air pressure inside an horizontal tube where air is being pushed through by two controllable fans at both ends. The goal is to infer the opening angle (in degrees) of a hatch $H \in [0, 45]$ on the side, given the pressure values inside the tunnel after applying a short power impulse to the intake fan. For the simulator, we use the model A2C3 from (Gamella et al., 2025, Appendix IV).

▶ Light Tunnel: In this task from Gamella et al. (2025), a camera is capturing light passing through two linear polarizers inside an elongated chamber. The goal is to predict RGB values of the light source and the polarizer effect $\alpha \in [0, 1]$, which is a function of the polarizer angle, so that $\boldsymbol{\theta} := [R, G, B, \alpha] \in [0, 255]^3 \times [0, 1]$. The simulator is a simplification of the real world process and described in (Gamella et al., 2025, Appendix IV) (Model F1). Observations are RGB images of size $(W, H, C) = (64, 64, 3)$ produced by either the simulator or the real apparatus. Model misspecification arises because the simulator omits certain physical effects present in the real measurements.

▶ Prior misspecification: We then also illustrate the impact of prior misspecification using two of the previous tasks, modifying the priors in the Gaussian and Wind Tunnel examples. Details are in Appendix D.4.

### 4.2. Evaluation Metrics and Calibration Diagnostics

$\mathcal{D}_{\text{test}}$ generally provides a single ground truth value $\boldsymbol{\theta}_j$ for each $\boldsymbol{y}_j$, so that metrics cannot be computed on parameter samples only as this would require multiple samples from

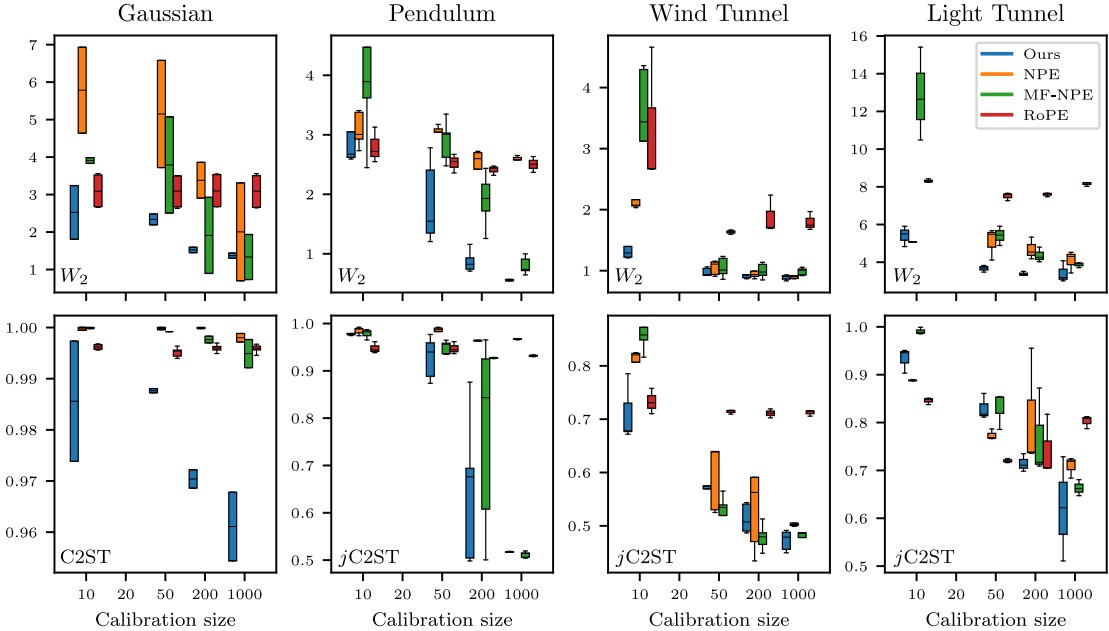

*Figure 2.* Wasserstein distance (top row, ↓ is better) and C2ST/$j$C2ST (bottom row, ↓ is better) with respect to an increasing calibration set size $N_{\text{cal}} \in \{10, 50, 200, 1000\}$. Each boxplot shows the distribution of metric values across five independent runs, each using a different randomly chosen calibration set. Note that for the Gaussian Task (left), the C2ST and $W_2$ distance are averaged over 3 different observations chosen at random.

the true posterior $p(\boldsymbol{\theta}|\boldsymbol{y}_j)$. Instead, for each $\boldsymbol{y}_j$ we generate one $\tilde{\boldsymbol{\theta}}_j$ from each learned posterior and compare $\mathcal{D}_{\text{test}}$ to the set of pairs $\{(\tilde{\boldsymbol{\theta}}_j, \boldsymbol{y}_j)\}_{1 \leq j \leq N_{\text{test}}}$ as samples from a joint distribution, except for the Gaussian task since the ground truth posterior distribution is available in this case. We assess the performance of our method performance based on the following four metrics.

**Classifier Two-Sample Test (C2ST) and joint C2ST ($j$C2ST).** The C2ST (Lopez-Paz & Oquab, 2017) measures the discrepancy between two distributions by training a binary classifier to distinguish samples drawn from them. The test statistic is the classifier's accuracy on samples from a test set, which is equal to chance-level (usually 0.5) for indistinguishable distributions and approaches 1 as the distributions diverge. The metric on parameter values used for the Gaussian example is then denoted by C2ST while for all other tasks we write $j$C2ST to indicate that the metric is computed over joint samples.

**Wasserstein Distance ($W_2$).** To provide a geometry-aware notion of distributional similarity, the Wasserstein distance for the $L_2$ cost is computed between samples from the approximate and true joint distribution $p(\boldsymbol{\theta}, \boldsymbol{y})$, except for the Gaussian task, unless otherwise specified.

$$W_2 = \min_{\gamma \in \mathbb{R}_+^{N_{\text{test}} \times N_{\text{test}}}} \left( \sum_{i,j} \gamma_{i,j} \|(\boldsymbol{\theta}_i, \boldsymbol{y}_i) - (\tilde{\boldsymbol{\theta}}_j, \boldsymbol{y}_j)\|_2^2 \right)^{\frac{1}{2}}$$

such that $\gamma.\mathbf{1} = \frac{1}{N_{\text{test}}}$ and $\gamma^T.\mathbf{1} = \frac{1}{N_{\text{test}}}$ .

**Maximum Mean Discrepancy** (MMD). We compute the Maximum Mean Discrepancy between samples from the true and approximate joint distributions using a radial basis function (RBF) kernel. Given samples $\{(\boldsymbol{\theta}_i, \boldsymbol{y}_i)\}_{1 \leq i \leq N_{\text{test}}}$ and $\{(\tilde{\boldsymbol{\theta}}_j, \boldsymbol{y}_j)\}_{1 \leq j \leq N_{\text{test}}}$, the squared MMD is

$$\text{MMD}^2 = \frac{1}{N_{\text{test}}(N_{\text{test}} - 1)} \sum_{i \neq i'} k_\sigma((\boldsymbol{\theta}_i, \boldsymbol{y}_i), (\boldsymbol{\theta}_{i'}, \boldsymbol{y}_{i'}))$$
$$+ \frac{1}{N_{\text{test}}(N_{\text{test}} - 1)} \sum_{j \neq j'} k_\sigma\left((\tilde{\boldsymbol{\theta}}_j, \boldsymbol{y}_j), (\tilde{\boldsymbol{\theta}}_{j'}, \boldsymbol{y}_{j'})\right)$$
$$- \frac{2}{N_{\text{test}}^2} \sum_{i,j} k_\sigma\left((\boldsymbol{\theta}_i, \boldsymbol{y}_i), (\tilde{\boldsymbol{\theta}}_j, \boldsymbol{y}_j)\right),$$

with $k_\sigma(x, x') = \exp\left(-\frac{\|x - x'\|_2^2}{2\sigma^2}\right)$ and $\sigma = 10$ in our experiments.

**Mean Squared Error (MSE).** The average mean squared error between $M$ generated samples $\{\tilde{\boldsymbol{\theta}}_j^i\}_{1 \leq i \leq M}$ and the ground truth parameters $\{\boldsymbol{\theta}_j\}_{1 \leq j \leq N_{\text{test}}}$ for each observation $\boldsymbol{y}_j$ in $\mathcal{D}_{\text{test}}$, is given by

$$\text{MSE} = \frac{1}{N_{\text{test}}} \frac{1}{M} \sum_{j}^{N_{\text{test}}} \sum_{i}^{M} \|\tilde{\boldsymbol{\theta}}_j^i - \boldsymbol{\theta}_j\|_2^2 .$$

MSE is a good accuracy measure for unimodal posteriors, which is often the case in our experiments.

**Posterior calibration diagnostics (SBC).** We also report the simulation-based calibration (SBC) diagnostic of Talts

et al. (2020). For each 1D marginals of the posterior, SBC provides an histogram whose deviation from uniformity is indicative of a potential issue in the posterior, possibly too narrow, too wide or skewed. In contrast, the absence of a significant deviation offers no guarantee that the posterior is correct, as uniformity trivially occurs when the posterior and the prior are the same. SBC requires $N$ samples from the true generative process $\{(\boldsymbol{\theta}_j, \boldsymbol{y}_j)\}_{1 \leq j \leq N}$ and $L$ independent draws from the estimated posteriors for each $\boldsymbol{y}_j$. The procedure is detailed in Appendix C. In practice, the sensitivity of SBC may be limited as true samples may only be available from a calibration data set of small size.

### 4.3. Results and discussion

Our analysis is detailed for the four tasks exhibiting simulator misspecifications. Additional experiments illustrating misspecifications in prior distributions are reported in Appendix D.4 with similar conclusions. Figure 2 and Appendix Figure 7, first illustrate the impact of $N_{\text{cal}}$ on posterior estimation, for the compared methods and various metrics. All metrics decrease as $N_{\text{cal}}$ increases, highlighting the importance of having enough calibration data to correct misspecification.

For a given ground truth couple $(\boldsymbol{\theta}^*, \boldsymbol{y}^*) \in \mathcal{D}_{\text{test}}$, Figure 3 illustrates the posterior density of the first two components of $\boldsymbol{\theta}$ for each method for the Gaussian and Pendulum tasks, except RoPE whose too flat posterior estimate hinders the other posterior distributions. Similar plots for the other tasks are provided in the Appendix in Figures 8 (Wind Tunnel) and 9 (Light Tunnel).

SBC diagnostics are reported in Appendix C, Figures 4 and 5, for our method and the other baselines.

**Comparison to baselines.** All experiments demonstrate that our method consistently outperforms all baselines, quantitatively and qualitatively. In Figure 2, we observe that for the Gaussian task, our approach achieves better performance than plain NPE, fine-tuned MFNPE, and RoPE. This gap becomes more striking in Figure 3 (second row), where both NPE and MFNPE produce multimodal posteriors even when the true posterior is unimodal. In contrast, our method yields unimodal posteriors that are better calibrated and centered on the ground-truth parameter $\boldsymbol{\theta}^*$, and this even for a low calibration size. For the Pendulum and Light Tunnel tasks, NPE fails to capture the intricate dependencies between parameters and observations—even when $N_{\text{cal}} = 1000$. In these cases, our method achieves superior performance in both $j$C2ST and $W_2$, while also exhibiting lower variance across draws of the calibration data set compared to MFNPE (Figure 2). Figure 3 further illustrates that MFNPE often produces overly sharp posterior distributions in one or more dimensions, yet fails to recover the true pa-

rameter $\boldsymbol{\theta}^*$. In contrast, our method consistently recovers $\boldsymbol{\theta}^*$ with higher accuracy. We hypothesize that this behavior stems from the MFNPE training procedure. MFNPE learns a neural encoder $h_\omega(\boldsymbol{x})$ to extract latent representations from simulated data $\boldsymbol{x}$. During fine-tuning, however, this encoder is evaluated on real observations $\boldsymbol{y}$, whose distribution differs from that of $\boldsymbol{x}$. This distributional shift leads to erroneous latent representations $h_\omega(\boldsymbol{y})$, and consequently to biased posteriors. While this issue diminishes as $N_{\text{cal}}$ increases, it remains pronounced at small calibration sizes ($N_{\text{cal}} = 10$ or $50$). The Pendulum task provides further evidence supporting this analysis. The misspecification arises from exponential damping, which predominantly affects the estimation of the oscillation amplitude $A$, as seen in Figure 3 (first row). In contrast, the estimation of the frequency $\omega_0$ remains accurate, since it is less sensitive to the damping mismatch. In the real-world tasks (Wind Tunnel and Light Tunnel), our approach yields substantially better results at small calibration sizes, particularly in terms of $W_2$, and remains competitive or superior for larger calibration sets. Appendix D provides additional visualizations of the posterior estimates for these tasks. The impact of our correction is also illustrated in Figures 10 to 13, comparing the posteriors before and after calibration.

SBC results show that our method is better calibrated as $N_{\text{cal}}$ grows and outperforms baselines such as NPE and MFNPE. For the Gaussian task, RoPE shows less deviation from uniformity at low $N_{\text{cal}}$ but with no significant improvement when $N_{\text{cal}}$ increases. We believe this is due to the regularization introduced by parameters $\gamma$ and $\rho$ in the OT formulation, which bias the RoPE posterior toward the prior.

**Ablation study.** To provide more insight on the different FMCPE components, we also discuss in Appendix D the choice of using a joint optimization scheme and compare FMCPE to other alternatives where we isolate each component of the method to ponder its effect on the final results. Figure 14 shows that FMCPE achieves better or comparable performances than the other methods being tested, in particular, it yields better results than training both flows in a sequential manner.

For completeness, we also conducted an experiment with a Gaussian model to check the effect of the conditional independence hypothesis (2) when we only use the mapping $T_{\boldsymbol{X}}$. Experiment details are reported in Appendix D.3. Results in Table 1 show that when the conditional hypothesis holds, using only $T_{\boldsymbol{X}}$ is sufficient to recover the ground truth posterior, but as more and more dependence between $\boldsymbol{\theta}$ and $\boldsymbol{y}$ is introduced in the model, results start to degrade.

**Training and Inference times.** Times for FMCPE and the baselines in Figure 2 are reported in Appendix E, Tables 2 and 3. FMCPE is more expensive to train but remains

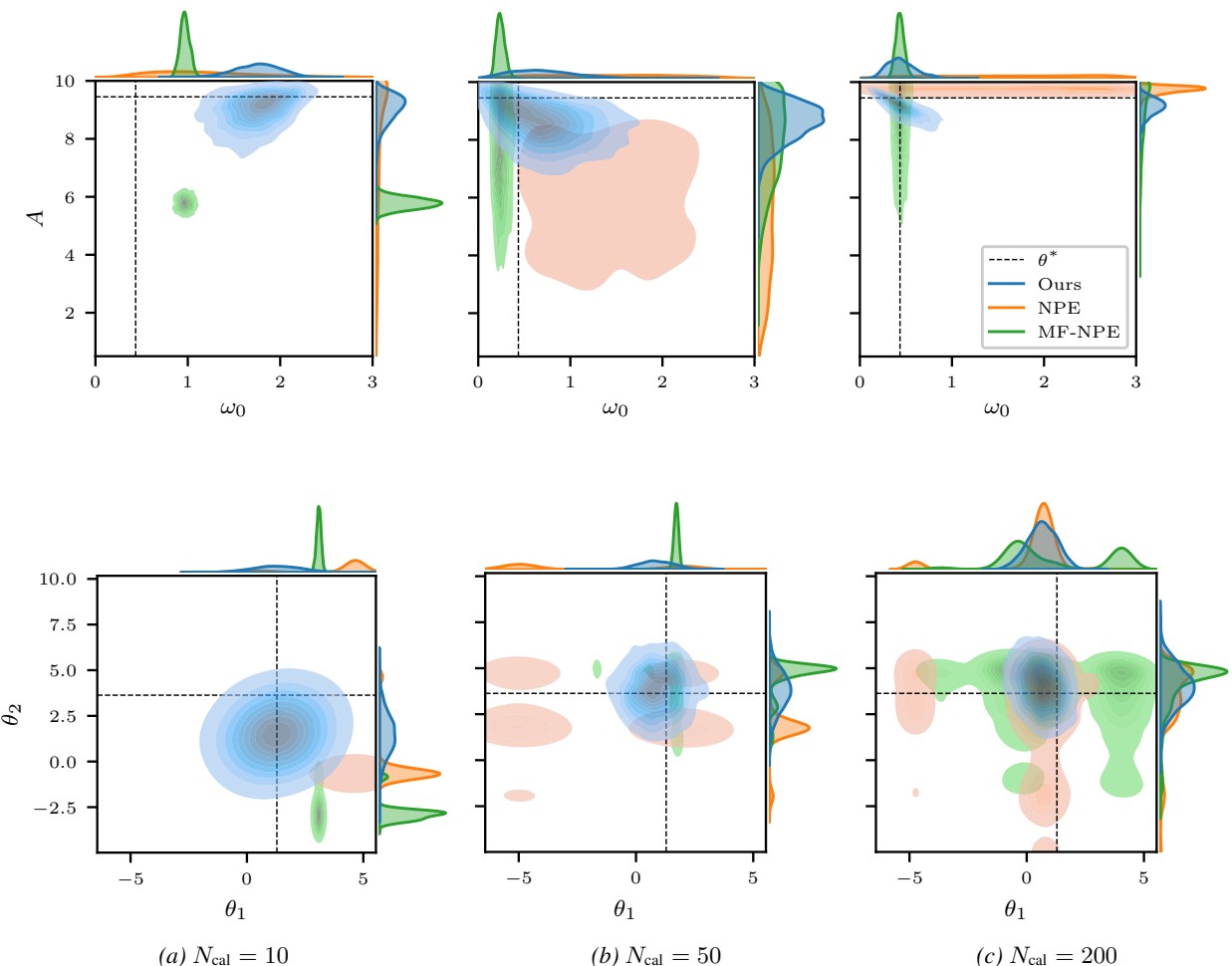

*Figure 3.* Kernel density estimates of joint and marginal samples for Pendulum (first row) and Gaussian (second row). For a given $\boldsymbol{y}^* \in \mathcal{D}_{\text{test}}$, we draw $\{\tilde{\boldsymbol{\theta}}_i\}_{1 \leq i \leq 2000}$, for each method and 3 calibration sizes $N_{\text{cal}} \in \{10, 50, 200\}$. Dotted black lines indicate the true parameter $\boldsymbol{\theta}^*$ that generated $\boldsymbol{y}^*$.

practical, requiring only 5–6 minutes per task, with space for speed up, as our current implementation does not optimize the flow sampling step. At inference time, FMCPE is faster than RoPE for all tasks and calibration sizes.

## 5. Conclusion

We tackled model misspecification in SBI by combining scarce calibration data with abundant simulations. Our method builds a proposal posterior from both sources and refines it via flow matching, producing posterior estimates that are more accurate and better calibrated than standard SBI baselines. Importantly, our proposal is also computationally efficient, as it can leverage off-the-shelf SBI posterior distributions as proposals, requiring only lightweight refinement with calibration data.

To complement these experimental observations, It would be interesting to further investigate whether useful bounds

could be derived to quantify the quality of our proposed correction and to possibly derive practical guidelines to set an effective calibration set size with respect to the task dimension and misspecification.

While severe misspecifications in high dimensions may still require larger calibration sets, our results show that even small amounts of real data can substantially improve inference quality. This highlights the promise of our framework as a practical and scalable way to bring SBI closer to real-world scientific applications, and opens exciting opportunities for richer proposal architectures, domain adaptation techniques, and deployment on large-scale simulators where misspecification is inevitable.

## Impact Statement

This paper presents a methodological contribution to simulation-based inference. Our method is not tied to any specific application domain and does not introduce new capabilities that would raise ethical or societal concerns beyond those already present in machine learning. We therefore do not anticipate significant negative societal impacts arising directly from this work.

## Acknowledgments

The authors would like to thank Antoine Wehenkel for his insights and helpful remarks on the RoPE algorithm. PLCR was supported by a national grant managed by the French National Research Agency (Agence Nationale de la Recherche) attributed to the SBI4C project of the MIAI AI Cluster, under the reference ANR-23-IACL-0006. MA was supported by the ANR project BONSAI (grant ANR-23-CE23-0012-01).

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

## A. Misspecification in SBI

For self-contentedness, we reproduce an example similar to that of Wehenkel et al. (2025) that shows that a simulator may be well-specified according to the standard definition of misspecification but still provide biased estimates of the target parameter when applied to real data.

Consider a process that produces observations $y_1, \ldots y_N$ of some quantity $\theta \in \mathbb{R}$ with a physical meaning of interest. The observations are realizations of a true DGP, $y \sim p_{\theta^*} = \mathcal{N}(\theta^*, \sigma^2)$ where $\theta^*$ is the true value. Assume that instead of the true DGP, we have access to a simulator which produces for a given $\theta$, simulations that are realizations of $x \sim p_\theta = \mathcal{N}(\theta + \lambda, \sigma^2)$ where $\lambda > 0$ is a fixed offset. According to the standard definition, we are in a well-specified setting as we are in an $\mathcal{M}$-closed problem since there exists a value of $\theta = \theta^* - \lambda$ such that $p_\theta = p_{\theta^*}$. However, the posterior estimates with such a simulator are biased with respect to $\theta^*$. To see that, let us assume a prior $\mathcal{N}(\theta_0, \sigma_0^2)$ on $\theta$. The specific choice of prior is not important as its impact vanishes as the number of observations $N$ tends to infinity but the above choice is convenient to use Bayesian conjugacy properties. Indeed, it follows that $p(\theta|y_1, \ldots, y_N)$ is Gaussian with,

$$p(\theta|y_1, \ldots, y_N) = \mathcal{N}(\mu_N, \sigma_N^2)$$

where $\sigma_N^2 = \left(\frac{1}{\sigma_0^2} + \frac{N}{\sigma^2}\right)^{-1}$ and $\mu_N = \sigma_N^2 \left(\frac{\theta_0}{\sigma_0^2} + \frac{1}{\sigma^2} \sum_{i=1}^{N}(y_i - \lambda)\right)$.

Therefore, when $N$ tends to infinity, $\mathbb{E}[\theta|y_1, \ldots, y_N] \to \mathbb{E}[y] - \lambda = \theta^* - \lambda$, showing a bias that does not vanish.

## B. Experimental Setup

We provide additional details on the training setup used across all experiments. For every task, we allocate a simulation budget of $N_{\text{sim}} = 5 \times 10^4$ samples and evaluate four calibration set sizes, $N_{\text{cal}} \in \{10, 50, 200, 1000\}$, which we denote by $N_1, N_2, N_3, N_4$.

During training, $20\%$ of each calibration set is reserved for validation and the remaining $80\%$ is used for training. We did not optimize over random seeds. All models were trained on a Nvidia RTX3060 GPU under 3 hours.

**Data preprocessing.** For tasks Pendulum, Wind Tunnel, and Light Tunnel (uniform priors), we apply a logit transformation to map prior samples into $\mathbb{R}^p$. All datasets are standardized (z-scored) prior to training.

**Calibration sets.** For each $N_{\text{cal}}$, we generate 5 calibration sets by subsampling from a larger pool of calibration data. To reduce variance across runs, the sets are constructed in a nested fashion: denoting by $\mathcal{D}_{N_r}^i$ the $i$-th calibration set of size $N_r$, we enforce $\mathcal{D}_{N_r}^i \subset \mathcal{D}_{N_s}^i$ whenever $r < s$.

**Neural Posterior Estimation (NPE).** For a fair comparison, the procedure implemented in RoPE is used for all compared methods. We rely on the UMNN flow implementation available at https://github.com/AWehenkel/UMNN. NPE is implemented with two components: a neural statistic estimator (NSE), $h_\omega(\mathbf{x})$, that encodes data into a low-dimensional representation, and a normalizing flow (NF) that maps a base distribution to the posterior. For task Gaussian, we use a standard neural spline flow (Durkan et al., 2019) and omit the NSE. For tasks Pendulum, Wind Tunnel and Light Tunnel, we reuse the architectures and hyperparameters from Wehenkel et al. (2025).

**Flow Matching.** We use the architecture of Wildberger et al. (2023) as a backbone. For the $\boldsymbol{\theta}$-space flow $u_{\boldsymbol{\Theta}}$, conditioning on $\mathbf{x}$ is implemented through a task-specific embedding network. For the data-space flow $u_{\boldsymbol{X}}$, we employ the same architecture but with a separate embedding head for $(\mathbf{x}_t, t)$, equipped with positional encoding.

**Evaluation details.** The Wasserstein distance is computed using the `POT` package with default settings. For the C2ST, we train a classifier based on an MLP backbone, augmented with an embedding network for $\mathbf{y}$; the embedding architecture matches the one used for the normalizing flow. We apply 3-fold cross-validation and report the average validation accuracy across folds. By default, we balance the two classes - $C = 0$ (true samples) and $C = 1$ (generated samples)—but note that stratified K-fold can also be used to handle class imbalance.

### B.1. Gaussian task

The Gaussian task is defined as

$$p_{\boldsymbol{\Theta}} = \mathcal{N}(\mu_\theta, \Sigma_\theta), \quad p_{\boldsymbol{X}|\boldsymbol{\Theta}} = \mathcal{N}(A\boldsymbol{\theta} + b, \Sigma_x), \quad p_{\boldsymbol{Y}|\boldsymbol{\Theta}} = \mathcal{N}(C\boldsymbol{\theta} + d, \Sigma_y), \tag{11}$$

where
$$\mu_\theta \in \mathbb{R}^3, \ \Sigma_\theta \in \mathbb{R}^{3\times3}, \ A \in \mathbb{R}^{10\times3}, \ b \in \mathbb{R}^{10}, \ \Sigma_x \in \mathbb{R}^{10\times10}, \ C \in \mathbb{R}^{10\times3}, \ d \in \mathbb{R}^{10}, \ \Sigma_y \in \mathbb{R}^{10\times10}.$$

All parameters above are drawn randomly at the start of the experiment. The exact values can be found in the code by using the same seed we used for the experiment, which is hard-coded in the main script.

### B.2. Pendulum task

We follow the setup of Wehenkel et al. (2025, Appendix I.2). We sample $N = 200$ timesteps $t_i \sim \mathcal{U}[0, 10]$ and define the simulator as

$$S: \quad \begin{aligned} \boldsymbol{\theta}, \epsilon = (\eta_1, \ldots, \eta_N, \varphi) &\longmapsto [x_1, \ldots, x_N]^T \\ \text{with } x_i &= A\cos(\omega_0 t_i + \varphi) + \eta_i, \end{aligned} \tag{12}$$

where $\varphi \sim \mathcal{U}[0, 2\pi]$, $\eta_i \sim \mathcal{N}(0, \sigma^2)$ with $\sigma = 0.1$ and $\boldsymbol{\theta} = [A, \omega_0]$ are the parameters of interest that we try to infer. The high-fidelity data-generating process (DGP) includes damping:

$$\text{DGP}: \quad \begin{aligned} \boldsymbol{\theta}, \epsilon = (\eta_1, \ldots, \eta_N, \varphi, \alpha) &\longmapsto [y_1, \ldots, y_N]^T \\ \text{with } y_i &= e^{-\alpha t_i} A\cos(\omega_0 t_i + \varphi) + \eta_i, \end{aligned} \tag{13}$$

where $\alpha \sim \mathcal{U}[0, 1]$, $\varphi \sim \mathcal{U}[0, 2\pi]$, $\eta_i \sim \mathcal{N}(0, \sigma^2)$. The prior is uniform: $p_\Theta = \mathcal{U}[0, 3] \times \mathcal{U}[0.5, 10]$.

### B.3. Wind Tunnel task

We use the `load_out_0.5_osr_downwind_4` experiment from the `wt_intake_impulse_v1` dataset (Gamella et al., 2025). The data consist of 50-step time series measuring air pressure in the chamber after an impulse applied to the input fan. A hatch on the side controls an additional opening, which can be controlled with precision. The inference task is to predict its position $H \in [0, 45]$. We adopt model A2C3 from the `causalchamber` package as the simulator.

### B.4. Light Tunnel task

We use the light tunnel experiment `uniform_ap_1.8_iso_500.0_ss_0.005` from the `lt_camera_v1` dataset (Gamella et al., 2025). A camera at the rear-end of an elongated chamber captures a light source emitted from the other end passing through two linear polarizers. We refer the reader to (Gamella et al., 2025) for more details about the mechanistic model of the tunnel. The inference task consists in inferring the color of the light source $((R, G, B) \in [0, 255]^3)$ as well as the Malus law (Collett, 2005) coefficient $\alpha \in [0, 1]$. The prior over these variables is uniform. The coefficient is a function of the polarizer angle $\alpha = \cos^2(\phi_1 - \phi_2)$, which are given in the dataset. The misspecification is introduced by omitting some physical aspects, which are detailed in (Gamella et al., 2025, Appendix D.IV.2.2).

## C. Posterior Calibration

Simulation-Based Calibration (SBC) (Talts et al., 2020) is a principled method for assessing the internal consistency of an approximation $\hat{p}(\boldsymbol{\theta} \mid \boldsymbol{y})$ to the true posterior. It exploits the self-consistency property of Bayesian inference: if the approximate posterior is exact, then parameters drawn from the prior should be uniformly distributed in rank among samples from the approximate posterior. Note however, that as many easy-to-deploy coverage procedures for SBI, SBC does not provide sufficient conditions for posterior correctness. In particular, if the approximate posterior always returns the prior distribution instead of an approximation to the posterior, then it passes the coverage diagnostic described below. Instead, SBC provides necessary conditions for posterior correctness, and thus enable detection of inaccuracies in the posterior estimate.

The procedure is as follows. For each pair $(\boldsymbol{\theta}_j, \boldsymbol{y}_j)$ from a test set $\mathcal{D}_N = \{(\boldsymbol{\theta}_j, \boldsymbol{y}_j)\}_{1 \le j \le N}$, we draw $L$ samples from the approximate posterior,

$$\{\tilde{\boldsymbol{\theta}}_j^{(l)}\}_{1 \le l \le L} \sim \hat{p}(\boldsymbol{\theta} \mid \boldsymbol{y}_j). \tag{14}$$

Since $\boldsymbol{\theta}$ is multivariate, calibration is assessed per coordinate. For each dimension $k \in \{1, \ldots, d\}$, we compute the marginal rank statistic

$$r_{j,k} = \sum_{l=1}^{L} \mathbf{1}\left[\tilde{\theta}_{j,k}^{(l)} < \theta_{j,k}\right] \in \{0, 1, \ldots, L\}, \tag{15}$$

where $\theta_{j,k}$ denotes the $k$-th component of $\boldsymbol{\theta}_j$. Under a well-calibrated posterior, the ranks $\{r_{j,k}\}_{1 \le j \le N}$ should be uniformly distributed over $\{0, \dots, L\}$ for every coordinate $k$. Systematic deviations from uniformity reveal specific failure modes: a U-shaped rank distribution indicates overdispersion (the approximate posterior is too wide), while a hump-shaped distribution indicates underdispersion (too narrow). A shift toward low or high ranks signals a global bias in the posterior mean.

In our experiments, we evaluate SBC using $N = 1000$ calibration pairs and $L = 500$ posterior samples per pair. We reported the results for both the Gaussian and Wind Tunnel in Figures 4 and 5 that show the respective histograms summarizing the ranks with a number of bins set to 20 using the rule recommended by Talts et al. (2020). Results are reported for a specific training seed, and are not averaged over multiple draws of the calibration set, unlike the metrics displayed in Section 4.3.

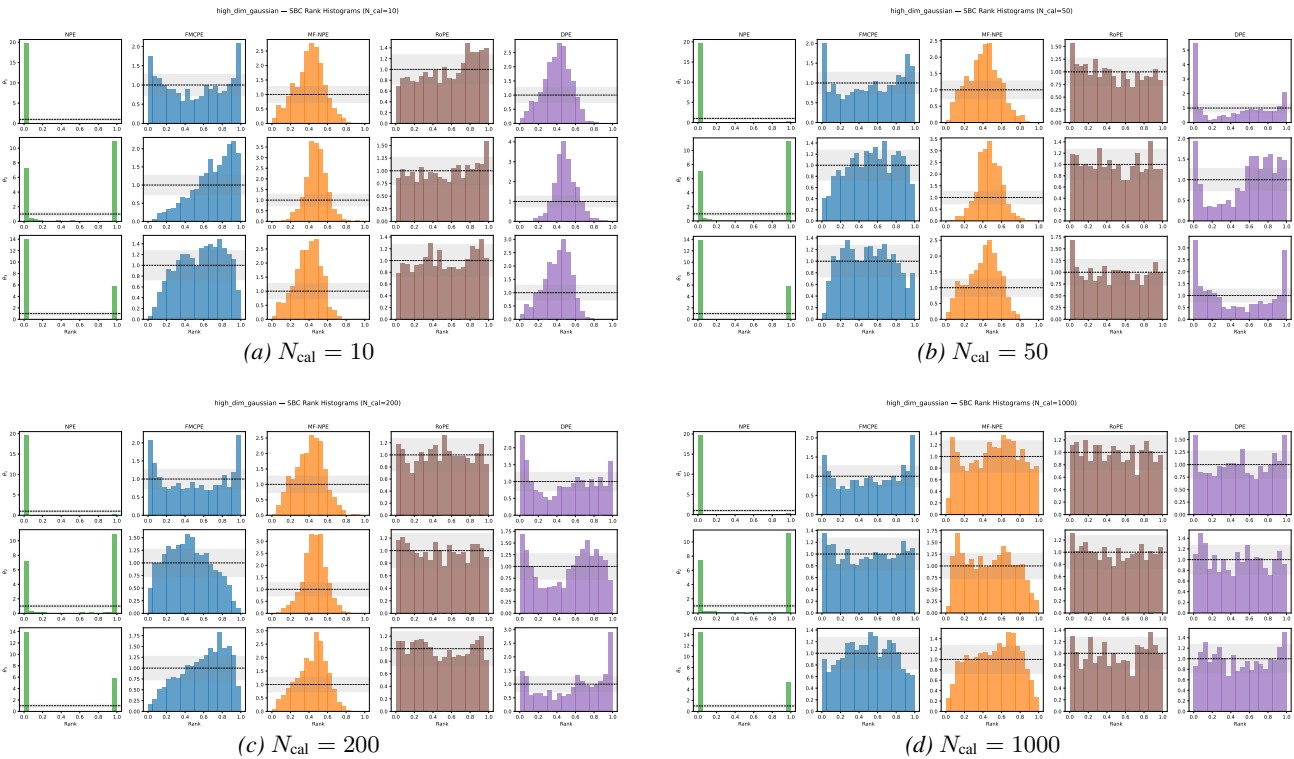

*(a) $N_{\text{cal}} = 10$*

*(b) $N_{\text{cal}} = 50$*

*(c) $N_{\text{cal}} = 200$*

*(d) $N_{\text{cal}} = 1000$*

*Figure 4.* Simulation Based Calibration for the Gaussian task. The four plots represent increasing $N_{\text{cal}}$ from 10 to 1000. In each plot, the rows correspond to the parameter dimensions $k = 1, 2, 3$. The histograms then show the distributions of the $\{r_{j,k}\}_{1 \le j \le N}$ for each method in columns, namely NPE, FMCPE, MFNPE, RoPE and DPE. A perfectly calibrated method should yield a uniform distribution, represented by the dashed black line. The gray band represents the 95% confidence interval of the bins distribution under the uniform assumption.

For the Gaussian task, all methods tend toward a uniform rank distribution at $N_{\text{cal}} = 1000$, except for the NPE baselines, which correspond to a NPE posterior trained solely on simulation data and evaluated on held-out calibration data. At lower values of $N_{\text{cal}}$, our method still yields satisfactory calibration compared to DPE and MFNPE, and is only outperformed by RoPE. We attribute this behavior to the implicit regularization introduced by the parameters $\rho$ and $\gamma$ in the OT problem, which bias the resulting posterior toward the prior and therefore naturally promote calibrated rank distributions.

For the Wind Tunnel task, all methods exhibit sub-optimal rank distributions even at $N_{\text{cal}} = 1000$. This may indicate either that the task is too challenging for the methods presented here to achieve satisfactory calibration, or that the SBC diagnostics were computed with insufficiently large values of $N$ and $L$. Since the observations are higher-dimensional than in the Gaussian task, we suspect that calibration could be better assessed given access to a larger number of test pairs $\{(\boldsymbol{\theta}_j, \boldsymbol{y}_j)\}_{1 \le j \le N}$.

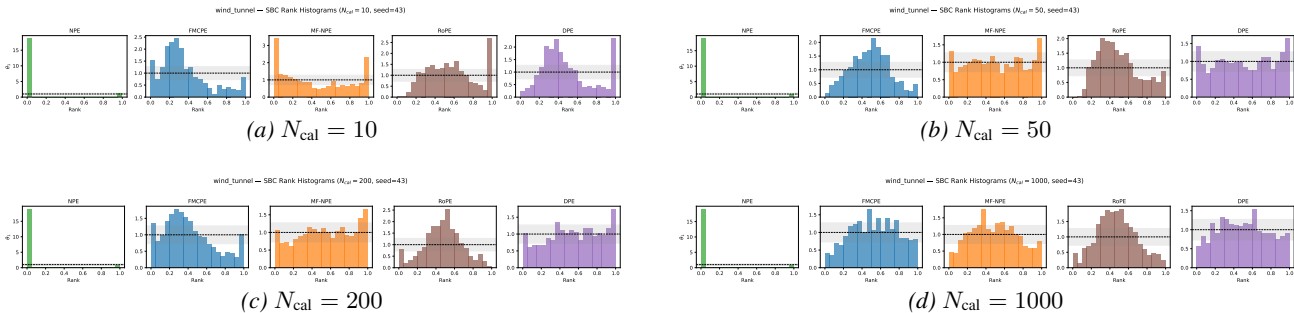

*Figure 5.* Simulation Based Calibration for the Wind Tunnel task. The four plots represent increasing $N_{\text{cal}}$ from 10 to 1000. In each plot, the histograms then show the distributions of the $\{r_j\}_{1 \leq j \leq N}$ for the single parameter and each method in columns, namely NPE, FMCPE, MFNPE, RoPE and DPE. A perfectly calibrated method should yield a uniform distribution, represented by the dashed black line. The gray band represents the 95% confidence interval of the bins distribution under the uniform assumption.

## D. Additional results

### D.1. Additional baselines and metrics

Additional metrics (MSE and MMD) are displayed for all previously compared methods, as well as for the following two additional baselines from Wehenkel et al. (2025),

- **Noisy Neural Posterior Estimation (NNPE)**: The amortized version of RNPE from Ward et al. (2022), where the authors added Spike and Slab noise to the simulations and then trained a NPE model on these noisy data. This allegedly makes the method more robust to model misspecification by providing more conservative posterior estimation.

- **Joint Neural Posterior Estimation (J-NPE):** A posterior estimator was trained on both simulated and calibration data jointly, $\mathcal{D}_{\text{train}} = \{(\boldsymbol{\theta}_i, \boldsymbol{x}_i)\}_{1 \leq i \leq N_{\text{sim}}} \bigcup \{(\boldsymbol{\theta}_j, \boldsymbol{y}_j)\}_{1 \leq j \leq N_{\text{cal}}}$.

As the ground truth posterior distribution is available for the Gaussian task, posterior-level metrics are computed and reported in Figure 6, namely,

- **C2ST**: The regular C2ST as introduced in (Lopez-Paz & Oquab, 2017).

- $\boldsymbol{W_2 - \theta}$: The Wasserstein distance between parameter samples $\{\tilde{\boldsymbol{\theta}}_i\}_{1 \leq i \leq M}$ from an estimated posterior $\tilde{p}(\boldsymbol{\theta}|\boldsymbol{y}^*)$ and samples $\{\boldsymbol{\theta}_i\}_{1 \leq i \leq M'}$ from the true posterior $p(\boldsymbol{\theta}|\boldsymbol{y}^*)$ for a given observation $\boldsymbol{y}^*$.

$$W_2 = \min_{\gamma \in \mathbb{R}_+^{M \times M'}} \left( \sum_{i,j} \gamma_{i,j} \|\boldsymbol{\theta}_i - \tilde{\boldsymbol{\theta}}_j\|_2^2 \right)^{\frac{1}{2}} \quad \text{such that } \gamma.\mathbf{1} = \frac{1}{M} \text{ and } \gamma^T.\mathbf{1} = \frac{1}{M'} .$$

- **MMD$-\theta$**: The Maximum Mean Discrepancy with a radial basis function (rbf) kernel between parameter samples $\{\tilde{\boldsymbol{\theta}}_i\}_{1 \leq i \leq M}$ from an estimated posterior $\tilde{p}(\boldsymbol{\theta}|\boldsymbol{y}^*)$ and samples $\{\boldsymbol{\theta}_i\}_{1 \leq i \leq M'}$ from the true posterior $p(\boldsymbol{\theta}|\boldsymbol{y}^*)$ for a given observation $\boldsymbol{y}^*$.

For all other tasks, metrics are computed between joint distribution samples and reported in Figure 7.

Additionally, KDE plots for tasks Wind Tunnel and Light Tunnel are shown, respectively, in Figures 8 and 9. RoPE was omitted in Figure 9 as it hindered clear visualization of the other posterior distributions due to a too flat posterior estimate.

Another qualitative illustration of the impact of our proposed correction is given in Figures 10 to 13. The plots clearly show how the posteriors are shifted to a more appropriate location after calibration.

### D.2. Ablation Study: impact of joint learning

To provide more insight on the different FMCPE components, an ablation study is also conducted. Our joint FMCPE training, referred to as $T_{\boldsymbol{\Theta}} + T_{\boldsymbol{X}}$ is first compared to a sequential one where the $T_{\boldsymbol{X}}$ map is first learned to provide the

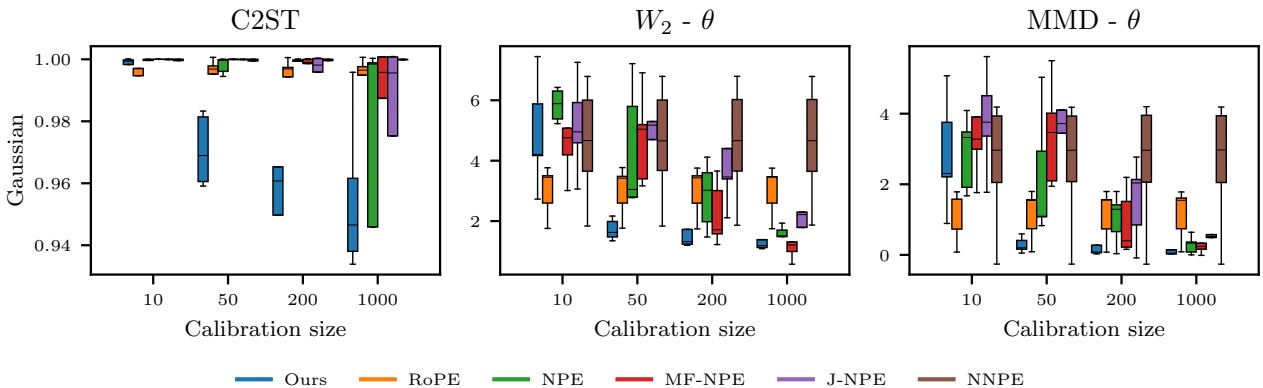

*Figure 6.* Parameter sample metrics ($\downarrow$ is better). C2ST, Wasserstein distance and Maximum Mean Discrepancy (MMD) for the Gaussian task. For a given observation $\boldsymbol{y}^*$, we sampled $M = 5000$ parameters $\{\tilde{\boldsymbol{\theta}}_i\}_{1 \leq i \leq M}$ for each method. Each boxplot show the distribution of metric values over 3 different observations $\boldsymbol{y}^* \in \mathcal{D}_{\text{test}}$ chosen at random.

source distribution for $T_{\boldsymbol{\Theta}}$ fitted in turns. Figure 14 shows that in all tested examples and metrics, the joint training provides better or equivalent performance, confirming the advantage of adjusting both maps in an interactive manner. To explore then the impact of each map separately, two options are considered. In the $T_{\boldsymbol{X}}$-only procedure, NPE is used with $\hat{p}_{\boldsymbol{\Theta}|\boldsymbol{X}}(\boldsymbol{\theta}|\boldsymbol{X} = T_{\boldsymbol{X}}(\boldsymbol{y}))$ where $T_{\boldsymbol{X}}$ is trained on calibration data, while $T_{\boldsymbol{\Theta}}$-only denotes a learned $T_{\boldsymbol{\Theta}}$ on calibration data from a source distribution $\pi(\boldsymbol{\theta}|\boldsymbol{y}) = \hat{p}_{\boldsymbol{\Theta}|\boldsymbol{X}}(\boldsymbol{\theta}|\boldsymbol{X} = \boldsymbol{y})$ ignoring calibration data. The worse performance of $T_{\boldsymbol{X}}$-only, including when calibration data increases, suggests that calibration has more impact when learning $T_{\boldsymbol{\Theta}}$ than $T_{\boldsymbol{X}}$. In contrast, $T_{\boldsymbol{\Theta}}$-only and $T_{\boldsymbol{\Theta}} + T_{\boldsymbol{X}} -$ Sequential, which differ mainly in the nature of the source, show similar performance. This suggests that a key feature of FMCPE is the possibility to learn jointly the source and $T_{\boldsymbol{\Theta}}$ rather than the nature of the source itself. Note that the Gaussian example is not shown because no significant differences were observed, which suggests that for simple models a sub-optimal use of calibration data may not be critical.

### D.3. Impact of the conditional independence hypothesis

The conditional independence (CI) hypothesis (Equation 2) is not a requirement for our method to work in general. We provide results on a toy Gaussian example where we control to which degree the CI holds with a scalar $\alpha$

$$\boldsymbol{\theta} \sim \mathcal{U}([-3,3]^5), \quad \boldsymbol{\theta} \in \mathbb{R}^5 \tag{16}$$

$$\boldsymbol{x} \mid \boldsymbol{\theta} \sim \mathcal{N}(A\boldsymbol{\theta}, \sigma_x^2 I), \quad \boldsymbol{x} \in \mathbb{R}^{20}, \ A \in \mathbb{R}^{20 \times 5} \tag{17}$$

$$\boldsymbol{y} \mid \boldsymbol{\theta}, \boldsymbol{x} \sim \mathcal{N}(C\boldsymbol{x} + \alpha D\boldsymbol{\theta}, \sigma_y^2 I), \quad C \in \mathbb{R}^{20 \times 20}, \ D \in \mathbb{R}^{20 \times 5} \tag{18}$$

where $\sigma_x = 0.5$, $\sigma_y = 0.3$. When $\alpha = 0$, $\boldsymbol{y}$ depends on $\theta$ only through $\boldsymbol{x}$ (CI holds); when $\alpha > 0$, the term $\alpha D\boldsymbol{\theta}$ introduces a direct $\boldsymbol{\theta} \to \boldsymbol{y}$ effect. We provide metrics values for FMCPE using both $T_{\boldsymbol{\Theta}}$ and $T_{\boldsymbol{X}}$ and FMCPE using only $T_{\boldsymbol{X}}$ in Table 1. We can clearly see that the gap between both variants is small for $\alpha = 0$, suggesting the approach using only $T_{\boldsymbol{X}}$ can be sufficient. The gap widens as $\alpha$ increases, confirming the positive impact of the $\boldsymbol{\theta}$-flow correction when conditional independence is violated.

### D.4. Prior Misspecification

We conducted an additional experiment to check how our method behaved when the source of misspecification comes from the prior. We considered the following modifications of two of the previous tasks:

- Gaussian task : We use a modified prior, which is shifted and scaled compared to the true prior $p(\boldsymbol{\theta})$.

$$p_{\text{miss}}(\boldsymbol{\theta}) = \mathcal{N}(\mu + \delta, \alpha\Sigma), \quad \text{while} \quad p(\boldsymbol{\theta}) = \mathcal{N}(\mu, \Sigma)$$

where $\mu \in \mathbb{R}^3$ and $\Sigma \in \mathbb{R}^{3 \times 3}$ are the mean and covariance of the Gaussian distribution. These are randomly generated one time and fixed for the rest of the experiment. Simulated data are drawn from $p_{\text{miss}}$ while calibration data use the true prior. The experiment was carried out with $\delta = 2.0$ and $\alpha = 3.0$. Results are shown in Figure 15.

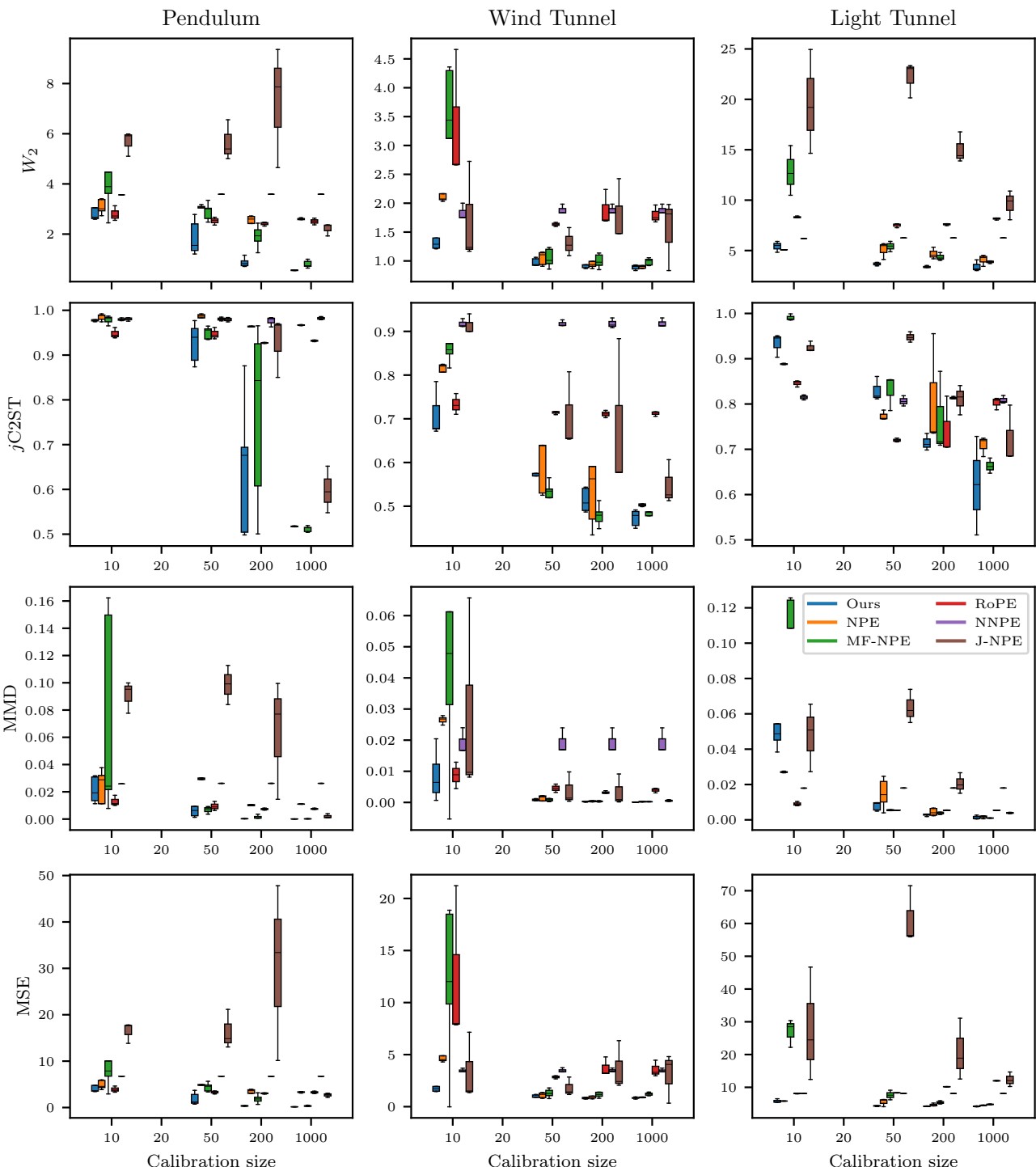

*Figure 7.* Joint sample metrics (↓ is better). Wasserstein distance (top row), $j$C2ST (second row) MMD (third row) and MSE (bottom row) with respect to an increasing calibration set size $N_{\text{cal}} \in \{10, 50, 200, 1000\}$. Each boxplot shows the distribution of metric values across five independent runs, each using a different randomly chosen calibration set.

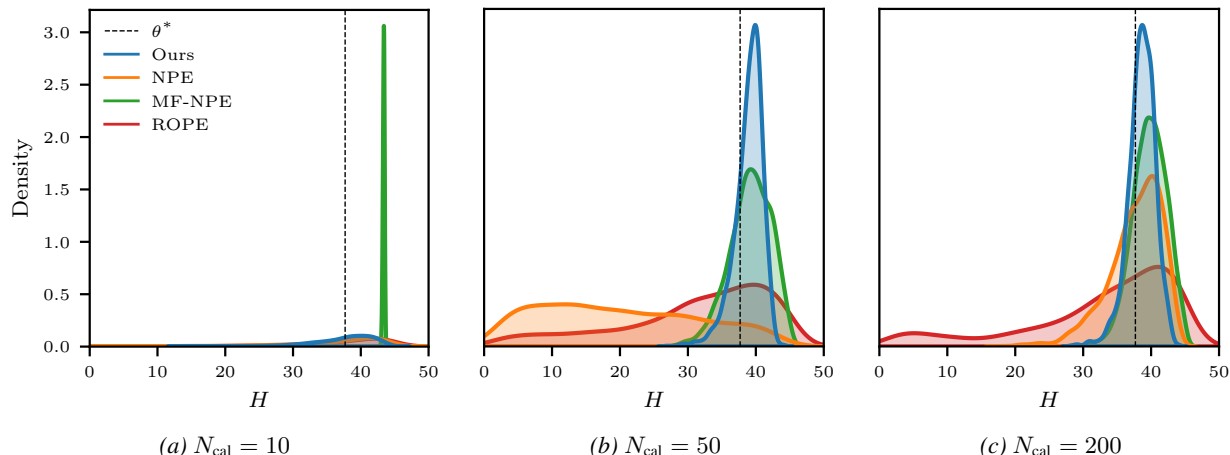

*Figure 8.* Kernel density estimates of the learned posteriors for task Wind Tunnel. For a given $\boldsymbol{y}^* \in \mathcal{D}_{\text{test}}$, we draw $\{\tilde{\boldsymbol{\theta}}_i\}_{1 \le i \le 2000}$, for each method and 3 calibration sizes $N_{\text{cal}} \in \{10, 50, 200\}$. The dotted black line indicates the true parameter $\boldsymbol{\theta}^*$ that generated $\boldsymbol{y}^*$.

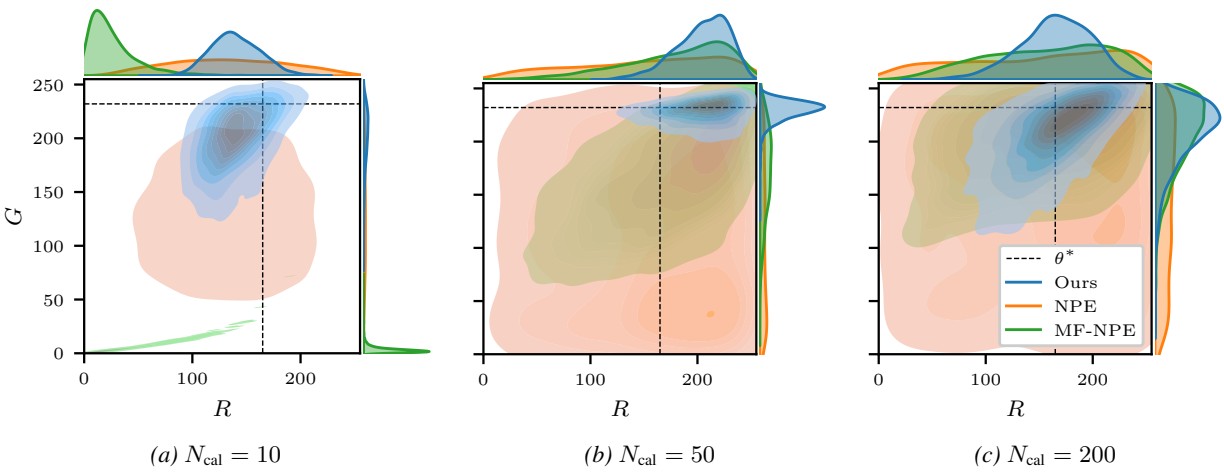

*Figure 9.* Kernel density estimates of joint and marginal samples for task Light Tunnel. We report the posterior densities for the first two coordinates of the parameter $[\theta_1, \theta_2] = (R, G)$. For a given $\boldsymbol{y}^* \in \mathcal{D}_{\text{test}}$, we draw $\{\tilde{\boldsymbol{\theta}}_i\}_{1 \le i \le 2000}$, for each method and 3 calibration sizes $N_{\text{cal}} \in \{10, 50, 200\}$. Dotted black lines indicate the true parameter $\boldsymbol{\theta}^*$ that generated $\boldsymbol{y}^*$.

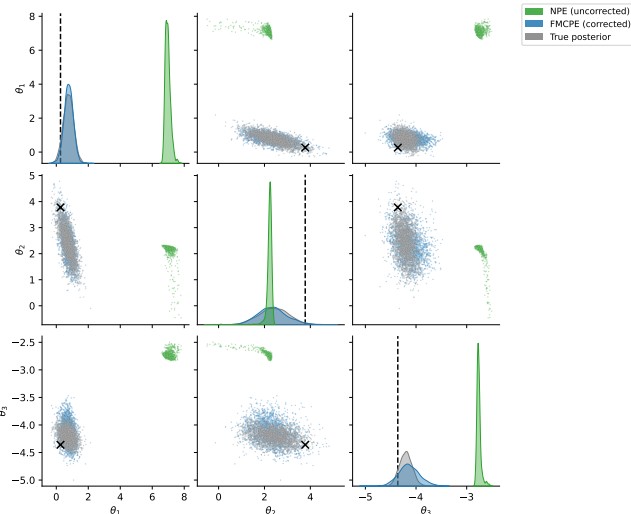

*Figure 10.* Gaussian task: for a given observation, posterior distributions before (NPE in green) and after our FMCPE correction (in blue) when $N_{\text{cal}} = 1000$. True posterior in grey.

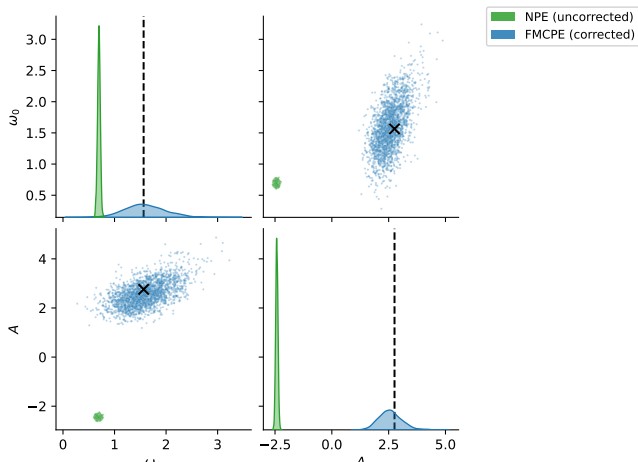

*Figure 11.* Pendulum task: for a given observation, posterior distributions before (NPE in green) and after our FMCPE correction (in blue) when $N_{\text{cal}} = 1000$.

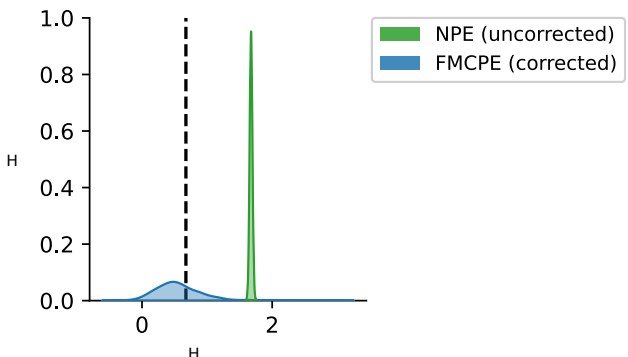

*Figure 12.* Wind Tunnel task: for a given observation, posterior distributions before (NPE in green) and after our FMCPE correction (in blue) when $N_{\text{cal}} = 1000$.

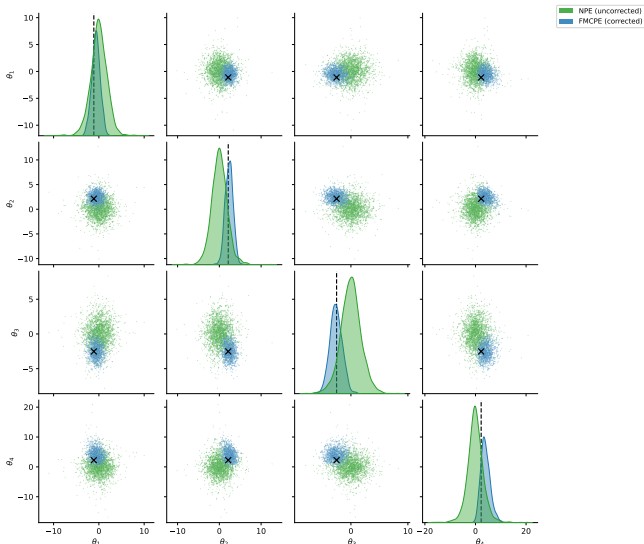

*Figure 13.* Light Tunnel task: for a given observation, posterior distributions before (NPE in green) and after our FMCPE correction (in blue) when $N_{\text{cal}} = 1000$.

| $\alpha$ | Method | jC2ST ($\downarrow 0.5$) | $\Delta\%$ | jMMD ($\downarrow 0$) | $\Delta\%$ |
|---|---|---|---|---|---|
| 0.0 | FMCPE (full) | $.576 \pm .006$ | | $.00040 \pm .00001$ | |
| | FMCPE ($T_{\boldsymbol{X}}$ only) | $.613 \pm .003$ | $+6.4\%$ | $.00043 \pm .00001$ | $+7.5\%$ |
| 0.5 | FMCPE (full) | $.630 \pm .015$ | | $.00045 \pm .00000$ | |
| | FMCPE ($T_{\boldsymbol{X}}$ only) | $.673 \pm .004$ | $+6.8\%$ | $.00051 \pm .00001$ | $+13.3\%$ |
| 1.0 | FMCPE (full) | $.691 \pm .016$ | | $.00030 \pm .00003$ | |
| | FMCPE ($T_{\boldsymbol{X}}$ only) | $.777 \pm .003$ | $+12.4\%$ | $.00048 \pm .00001$ | $+60.0\%$ |
| 2.0 | FMCPE (full) | $.768 \pm .041$ | | $.00022 \pm .00003$ | |
| | FMCPE ($T_{\boldsymbol{X}}$ only) | $.864 \pm .004$ | $+12.5\%$ | $.00045 \pm .00001$ | $+104.5\%$ |
| 5.0 | FMCPE (full) | $.545 \pm .051$ | | $.00033 \pm .00017$ | |
| | FMCPE ($T_{\boldsymbol{X}}$ only) | $.709 \pm .168$ | $+30.1\%$ | $.00052 \pm .00008$ | $+57.6\%$ |

*Table 1.* Flow-$\boldsymbol{\theta}$ ablation, $N_{\text{cal}} = 1000$. Mean $\pm$ std over 3 seeds. The gap between FMCPE (full) and FMCPE ($T_{\boldsymbol{X}}$ only) is reported relatively to the value of FMCPE (full) for each line.

- Wind Tunnel task : For this task, the true prior is set to an uniform distribution $p(\boldsymbol{\theta}) = \mathcal{U}[0, 45]$ while the misspecified prior is set to a mixture between the true prior and a narrower uniform distribution

$$p_{\text{miss}} = (1 - \tau)\,\mathcal{U}[0, 45]) + \tau\,\mathcal{U}[10, 20]$$

where $\tau$ is set to $\tau = 1.0$. Other values for $\tau$ have been considered and yielded comparable results and where not included here. We used the same simulator (model A2C3) to generate the simulation and calibration data. Results are shown in Figure 16.

For both tasks, our method gets better as $N_{\text{cal}}$ increases and outperforms the two other tested baselines, MFNPE and RoPE. RoPE does not seem to adapt to this particular type of misspecification, its performance does not increase with the size of the calibration set.

## E. Training and inference times

We report the training (Table 2) and inference (Table 3) times for our method and the baselines compared in Figure 2. Table 4 provides a detailed breakdown of the training time of our method across its main components. Although our method is more

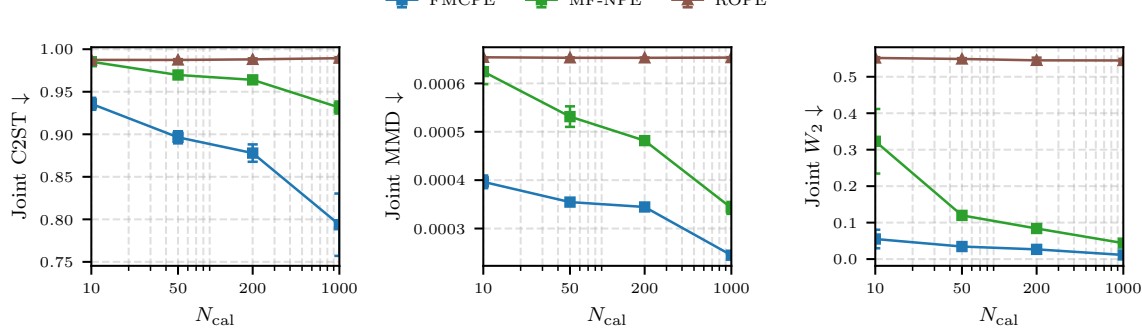

*Figure 14.* Ablation study investigating the impact of each FMCPE component (the lower the better). Our joint $T_\Theta + T_X$ (blue) is compared to a sequential version (orange) learning $T_X$ then $T_\Theta$, and to two other options. In $T_X$ only (green), resp. $T_\Theta$ only (red), calibration data is used only in learning $T_X$, resp. $T_\Theta$.

*Figure 15.* Method comparison under prior misspecification on the Gaussian task. Joint C2ST, MMD, and W2 between each method's posterior samples and the reference posterior, with respect to the calibration set size . Means $\pm$ standard deviations are reported across 3 draws for the calibration set.

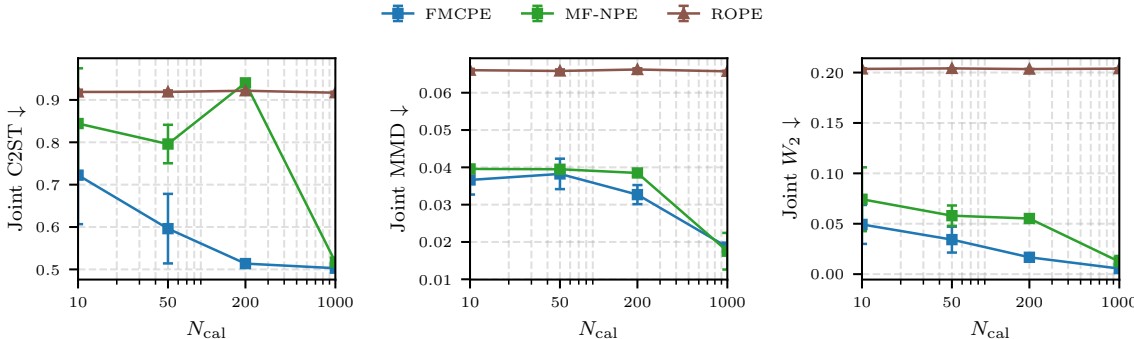

*Figure 16.* Method comparison under the most severe prior misspecification ($\tau = 1$) on the Wind Tunnel task. Joint C2ST, MMD, and W2 between each method's posterior samples and the reference posterior, with respect to calibration size. Means $\pm$ standard deviations are reported across 3 draws of the calibration set

expensive to train than the baselines, it remains practical, requiring only 5–6 minutes per task. As shown in Table 4, most of the training cost arises from the flow-sampling step (ODE integration), which is not optimized in our current implementation. We use a basic explicit-Euler solver, whereas more advanced integrators (e.g., midpoint methods, higher-order adaptive solvers, or JAX-based JIT compilation) could significantly reduce the number of integration steps and thus speed up training. Other methods such as distillation (Dao et al., 2025) or straight flow trajectories (Kornilov et al., 2024) can help us reduce the training time even further. At inference time, our method is faster than RoPE for all tasks and calibration sizes.

## F. Hyperparameter sensitivity

We conduct a grid search over the learning rate $\eta$, with $\eta \in \{10^{-4}, 5 \times 10^{-4}, 10^{-3}, 5 \times 10^{-3}\}$, and over the gradient clipping threshold $g$, with $g \in \{0.5, 1.0, 5.0, \text{none}\}$, to assess the sensitivity of our method to these two commonly tuned hyperparameters. All other hyperparameters are fixed, we chose $N_{\text{epochs}} = 50$ epochs and batch size $B = 256$. We fixed $N_{\text{cal}} = 200$, $N_{\text{sim}} = 50{,}000$, $N_{\text{test}} = 3{,}000$ and ran the grid search on 3 seeds $\in \{33, 43, 53\}$. We display a summary of the results in Table 5 for the Pendulum task and in Table 6 for the Light Tunnel task. We also provide a visual representation of each metric sensitivity to the hyperparameters in Figures 17 and 18.

The metrics used are the average joint Wasserstein-2 ($jW_2$, $\downarrow$) and Joint MMD ($\downarrow$) distances $\pm$ std over 3 seeds. On the Pendulum task, performance varies meaningfully with the learning rate: $\eta = 10^{-3}$ yields the best results on both metrics, with $jW_2$ dropping by a factor of 2.5 compared to the slowest rate $\eta = 10^{-4}$. The highest rate $\eta = 5 \times 10^{-3}$ leads to increased variance across seeds, suggesting mild instability. In contrast, gradient clipping has little effect across all learning rates. The four lines in Figure 17 nearly overlap, indicating that the method is robust to this hyperparameter.

On the Light Tunnel task, $jW_2$ decreases more gradually and monotonically as the learning rate increases, with no clear optimum within the tested range. The jMMD metric is essentially flat across all 16 configurations, varying by less than 0.2%, which confirms that the method is highly stable on this task regardless of the hyperparameter choices made.

Overall, the learning rate has a moderate effect on $jW_2$ in both tasks, while gradient clipping appears largely inconsequential. We select $\eta = 10^{-3}$, $g = 1.0$ as our default configuration based on the Pendulum results.

*Table 2.* Training time (in seconds) for each method and calibration set size $N_{cal}$. For the Gaussian task, no training time is reported for RoPE as no embedding network is used for NPE.

| Method | Gaussian | Light Tunnel | Pendulum | Wind Tunnel |
|---|---|---|---|---|
| | | $N_{cal} = 10$ | | |
| NPE | 5.51 | 0.79 | 0.63 | 0.58 |
| Ours | 28.35 | 37.12 | 52.82 | 94.73 |
| MF-NPE | 3.48 | 0.55 | 0.72 | 1.53 |
| RoPE | – | 5.40 | 7.81 | 7.10 |
| | | $N_{cal} = 50$ | | |
| NPE | 6.07 | 0.71 | 0.26 | 0.14 |
| Ours | 63.29 | 70.97 | 80.74 | 126.74 |
| MF-NPE | 6.33 | 1.76 | 0.76 | 1.92 |
| RoPE | – | 6.53 | 7.95 | 7.67 |
| | | $N_{cal} = 200$ | | |
| NPE | 13.11 | 4.11 | 2.21 | 0.19 |
| Ours | 70.24 | 136.18 | 163.91 | 127.65 |
| MF-NPE | 10.35 | 4.38 | 2.86 | 3.02 |
| RoPE | – | 15.93 | 11.87 | 11.73 |
| | | $N_{cal} = 1000$ | | |
| NPE | 37.32 | 15.10 | 12.32 | 5.94 |
| Ours | 141.96 | 306.55 | 333.70 | 235.78 |
| MF-NPE | 22.33 | 26.76 | 8.45 | 10.99 |
| RoPE | – | 61.57 | 38.09 | 38.47 |

*Table 3.* Inference time (in seconds) for each method and calibration set size $N_{\text{cal}}$. For each method and task, we evaluated a test set of $N_{\text{test}} = 5000$ real-world observations $\{y_j\}_{1 \leq j \leq N_{\text{test}}}$. The reported times correspond to the total duration to generate all 5000 corresponding parameters.

| Method | Gaussian | Light Tunnel | Pendulum | Wind Tunnel |
|---|---|---|---|---|
| | | $N_{\text{cal}} = 10$ | | |
| NPE | 0.09 | 0.67 | 0.12 | 0.02 |
| Ours | 0.64 | 10.95 | 2.20 | 0.38 |
| MF-NPE | 0.01 | 0.69 | 0.06 | 0.01 |
| RoPE | 110.31 | 222.51 | 446.48 | 54.80 |
| | | $N_{\text{cal}} = 50$ | | |
| NPE | 0.01 | 0.63 | 0.07 | 0.01 |
| Ours | 0.73 | 10.81 | 2.22 | 0.41 |
| MF-NPE | 0.01 | 0.70 | 0.06 | 0.01 |
| RoPE | 106.94 | 218.01 | 93.07 | 54.32 |
| | | $N_{\text{cal}} = 200$ | | |
| NPE | 0.01 | 0.64 | 0.07 | 0.01 |
| Ours | 0.55 | 10.83 | 2.21 | 0.43 |
| MF-NPE | 0.01 | 0.58 | 0.08 | 0.01 |
| RoPE | 101.60 | 224.80 | 87.91 | 56.14 |
| | | $N_{\text{cal}} = 1000$ | | |
| NPE | 0.01 | 0.64 | 0.06 | 0.01 |
| Ours | 0.40 | 11.03 | 2.20 | 0.43 |
| MF-NPE | 0.01 | 0.65 | 0.08 | 0.01 |
| RoPE | 100.11 | 226.29 | 88.23 | 59.13 |

*Table 4.* Training time (in seconds) for our method (FMCPE) broken down into 3 components: (1) Gradient Step: time to perform gradient updates on the calibration set; (2) Sample NPE: time to sample from the base NPE model; (3) Sample $x$-flow: time to sample from the learned flow in the observation space.

| Task | Component | $N_{\text{cal}} = 10$ | $N_{\text{cal}} = 50$ | $N_{\text{cal}} = 200$ | $N_{\text{cal}} = 1000$ |
|---|---|---|---|---|---|
| Gaussian | Gradient Step | 0.18 | 0.68 | 0.84 | 1.67 |
| | Sample NPE | 8.96 | 19.84 | 21.70 | 43.54 |
| | Sample $x$-flow | 19.20 | 42.76 | 47.70 | 96.75 |
| Light Tunnel | Gradient Step | 0.68 | 1.43 | 2.36 | 2.81 |
| | Sample NPE | 9.82 | 18.24 | 33.68 | 58.25 |
| | Sample $x$-flow | 26.62 | 51.29 | 100.14 | 245.49 |
| Pendulum | Gradient Step | 0.94 | 1.86 | 3.74 | 5.97 |
| | Sample NPE | 11.04 | 16.75 | 34.45 | 70.04 |
| | Sample $x$-flow | 40.83 | 62.12 | 125.72 | 257.69 |
| Wind Tunnel | Gradient Step | 1.10 | 1.68 | 1.84 | 2.90 |
| | Sample NPE | 21.03 | 27.93 | 27.91 | 52.04 |
| | Sample $x$-flow | 72.60 | 97.13 | 97.90 | 180.83 |

*Table 5.* Hyperparameter sensitivity (learning rate $\eta$ and gradient clipping $g$) on the Pendulum task ($N_{\text{cal}} = 200$, mean $\pm$ std over seeds $\{33, 43, 53\}$). Best result per metric highlighted in bold **green**.

| $\eta$ | $g$ | $jW_2 \downarrow$ | jMMD $\downarrow$ |
|---|---|---|---|
| $10^{-4}$ | 0.5 | $6.43 \times 10^{-3}_{\pm 0.14 \times 10^{-3}}$ | $7.32 \times 10^{-4}_{\pm 0.07 \times 10^{-4}}$ |
| | 1.0 | $6.45 \times 10^{-3}_{\pm 0.15 \times 10^{-3}}$ | $7.37 \times 10^{-4}_{\pm 0.07 \times 10^{-4}}$ |
| | 5.0 | $6.38 \times 10^{-3}_{\pm 0.16 \times 10^{-3}}$ | $7.36 \times 10^{-4}_{\pm 0.11 \times 10^{-4}}$ |
| | none | $6.39 \times 10^{-3}_{\pm 0.14 \times 10^{-3}}$ | $7.35 \times 10^{-4}_{\pm 0.06 \times 10^{-4}}$ |
| $5 \times 10^{-4}$ | 0.5 | $5.80 \times 10^{-3}_{\pm 0.39 \times 10^{-3}}$ | $7.28 \times 10^{-4}_{\pm 0.03 \times 10^{-4}}$ |
| | 1.0 | $5.71 \times 10^{-3}_{\pm 0.41 \times 10^{-3}}$ | $7.27 \times 10^{-4}_{\pm 0.04 \times 10^{-4}}$ |
| | 5.0 | $5.85 \times 10^{-3}_{\pm 0.37 \times 10^{-3}}$ | $7.28 \times 10^{-4}_{\pm 0.04 \times 10^{-4}}$ |
| | none | $5.82 \times 10^{-3}_{\pm 0.37 \times 10^{-3}}$ | $7.28 \times 10^{-4}_{\pm 0.10 \times 10^{-4}}$ |
| $10^{-3}$ | 0.5 | $2.66 \times 10^{-3}_{\pm 0.41 \times 10^{-3}}$ | $6.97 \times 10^{-4}_{\pm 0.11 \times 10^{-4}}$ |
| | 1.0 | $\mathbf{2.60 \times 10^{-3}}_{\pm 0.52 \times 10^{-3}}$ | $\mathbf{6.93 \times 10^{-4}}_{\pm 0.16 \times 10^{-4}}$ |
| | 5.0 | $2.71 \times 10^{-3}_{\pm 0.33 \times 10^{-3}}$ | $6.95 \times 10^{-4}_{\pm 0.18 \times 10^{-4}}$ |
| | none | $2.75 \times 10^{-3}_{\pm 0.30 \times 10^{-3}}$ | $7.02 \times 10^{-4}_{\pm 0.16 \times 10^{-4}}$ |
| $5 \times 10^{-3}$ | 0.5 | $4.53 \times 10^{-3}_{\pm 1.63 \times 10^{-3}}$ | $7.34 \times 10^{-4}_{\pm 0.09 \times 10^{-4}}$ |
| | 1.0 | $6.44 \times 10^{-3}_{\pm 0.58 \times 10^{-3}}$ | $7.44 \times 10^{-4}_{\pm 0.13 \times 10^{-4}}$ |
| | 5.0 | $4.76 \times 10^{-3}_{\pm 1.72 \times 10^{-3}}$ | $7.37 \times 10^{-4}_{\pm 0.09 \times 10^{-4}}$ |
| | none | $3.85 \times 10^{-3}_{\pm 1.49 \times 10^{-3}}$ | $7.40 \times 10^{-4}_{\pm 0.08 \times 10^{-4}}$ |

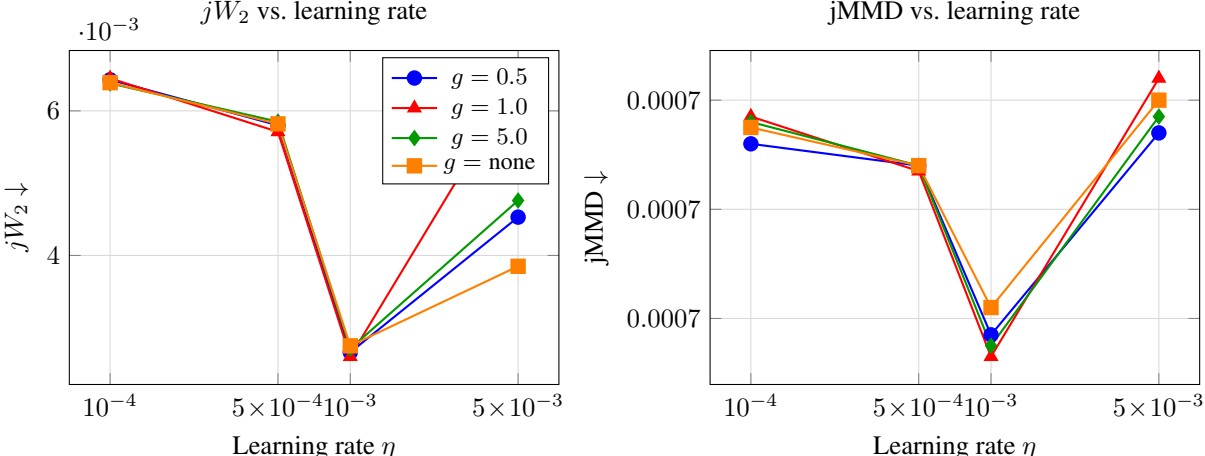

*Figure 17.* Hyperparameter sensitivity for the Pendulum task. Joint $W_2$ and MMD vs. learning rate. Lines for different $g$ nearly overlap, demonstrating robustness with respect to gradient clipping.

*Table 6.* Hyperparameter sensitivity on the Light Tunnel task ($N_{\text{cal}} = 200$, mean $\pm$ std over seeds $\{33, 43, 53\}$). Best result per metric in bold **green**.

| $\eta$ | $g$ | $jW_2 \downarrow$ | jMMD $\downarrow$ |
|---|---|---|---|
| $10^{-4}$ | 0.5 | $5.29 \times 10^{-3}{}_{\pm 0.43 \times 10^{-3}}$ | $6.466 \times 10^{-4}{}_{\pm 0.004 \times 10^{-4}}$ |
| | 1.0 | $5.26 \times 10^{-3}{}_{\pm 0.45 \times 10^{-3}}$ | $6.466 \times 10^{-4}{}_{\pm 0.006 \times 10^{-4}}$ |
| | 5.0 | $5.21 \times 10^{-3}{}_{\pm 0.54 \times 10^{-3}}$ | $6.466 \times 10^{-4}{}_{\pm 0.006 \times 10^{-4}}$ |
| | none | $5.30 \times 10^{-3}{}_{\pm 0.50 \times 10^{-3}}$ | $6.467 \times 10^{-4}{}_{\pm 0.007 \times 10^{-4}}$ |
| $5 \times 10^{-4}$ | 0.5 | $5.01 \times 10^{-3}{}_{\pm 0.65 \times 10^{-3}}$ | $6.474 \times 10^{-4}{}_{\pm 0.004 \times 10^{-4}}$ |
| | 1.0 | $5.04 \times 10^{-3}{}_{\pm 0.60 \times 10^{-3}}$ | $6.476 \times 10^{-4}{}_{\pm 0.004 \times 10^{-4}}$ |
| | 5.0 | $4.92 \times 10^{-3}{}_{\pm 0.56 \times 10^{-3}}$ | $6.474 \times 10^{-4}{}_{\pm 0.005 \times 10^{-4}}$ |
| | none | $4.96 \times 10^{-3}{}_{\pm 0.67 \times 10^{-3}}$ | $6.475 \times 10^{-4}{}_{\pm 0.005 \times 10^{-4}}$ |
| $10^{-3}$ | 0.5 | $4.42 \times 10^{-3}{}_{\pm 0.15 \times 10^{-3}}$ | $6.472 \times 10^{-4}{}_{\pm 0.004 \times 10^{-4}}$ |
| | 1.0 | $4.38 \times 10^{-3}{}_{\pm 0.11 \times 10^{-3}}$ | $6.472 \times 10^{-4}{}_{\pm 0.003 \times 10^{-4}}$ |
| | 5.0 | $4.46 \times 10^{-3}{}_{\pm 0.12 \times 10^{-3}}$ | $6.472 \times 10^{-4}{}_{\pm 0.004 \times 10^{-4}}$ |
| | none | $4.40 \times 10^{-3}{}_{\pm 0.20 \times 10^{-3}}$ | $6.471 \times 10^{-4}{}_{\pm 0.004 \times 10^{-4}}$ |
| $5 \times 10^{-3}$ | 0.5 | $4.06 \times 10^{-3}{}_{\pm 0.98 \times 10^{-3}}$ | $6.471 \times 10^{-4}{}_{\pm 0.009 \times 10^{-4}}$ |
| | 1.0 | $4.03 \times 10^{-3}{}_{\pm 0.98 \times 10^{-3}}$ | $6.471 \times 10^{-4}{}_{\pm 0.008 \times 10^{-4}}$ |
| | 5.0 | $4.51 \times 10^{-3}{}_{\pm 0.62 \times 10^{-3}}$ | $6.467 \times 10^{-4}{}_{\pm 0.012 \times 10^{-4}}$ |
| | none | $\mathbf{3.78 \times 10^{-3}}{}_{\pm 1.18 \times 10^{-3}}$ | $\mathbf{6.468 \times 10^{-4}}{}_{\pm 0.009 \times 10^{-4}}$ |

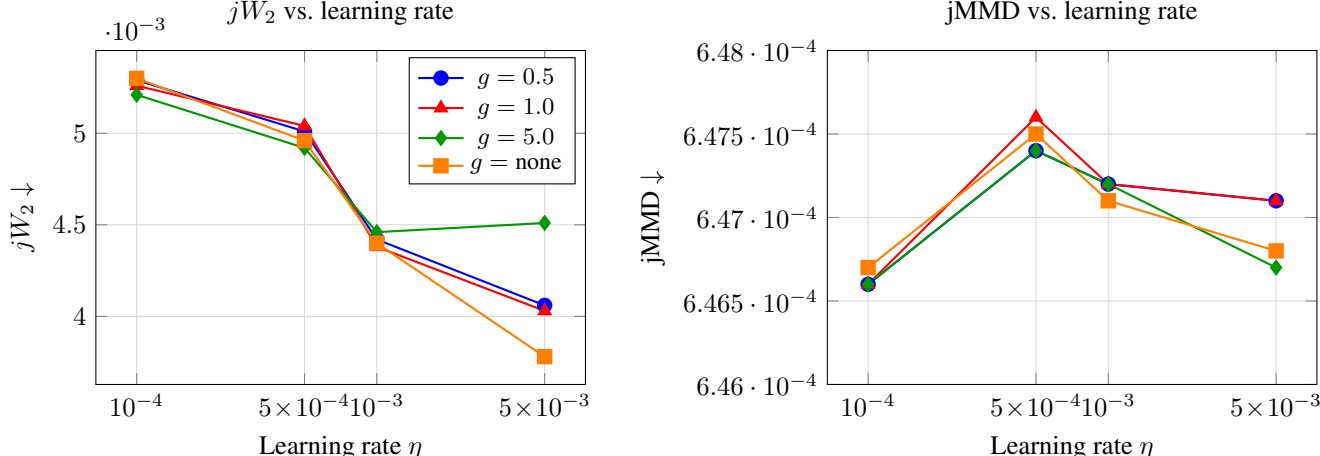

*Figure 18.* Hyperparameter sensitivity for the Light Tunnel task: Joint $W_2$ and MMD vs. learning rate. $jW_2$ decreases monotonically with $\eta$ on this task. Joint MMD is essentially constant across all 16 configurations (range $< 0.2\%$), confirming extreme robustness.

