# OpenReview forum: "Flow Matching Calibration for Simulation-Based Inference under Model Misspecification"
_ICML.cc/2026/Conference — ICML 2026 regular_

### Official Review · Reviewer_uvU7 · 2026-03-06

**Soundness:** 3
**Presentation:** 3
**Significance:** 3
**Originality:** 2
**Overall Recommendation:** 5
**Confidence:** 4

**Summary:**

The manuscript studies simulation-based inference (SBI) under model misspecification specifically in cases where the simulator may be biased relative to the "true" generative process of data. The manuscript proposes, FMCPE, which uses *two* flow matching stages to augment a pre-trained NPE estimator on misspecified/biased simulation data. The first flow matching map acts in data-space and transport "real" observations into the "simulation" data space. The second flow matching map acts in parameter-space and transports the biased NPE towards the true posterior. These are trained on a small "calibration set" (on "real"/unbiased theta, x pairs), while NPE trained on biased simulations is kept frozen. Empirically the paper demonstrate the effectiveness of this approach compared to RoPE (only in X-space) and MF-NPE (refine pretrained NPE) and report generally superior performance, especially with a small set of calibration samples.

**Compliance With Llm Reviewing Policy:**

Affirmed.

**Final Justification:**

My main concerns regarding some of the manuscript "side claims" have been resolved (also with additional experiments). I hence increased my score.

**Key Questions For Authors:**

- Can the authors clarify for each task if $\theta$ and $y$ are conditionally independet?
- As far as I understood a main contribution of the manuscript is to indirectly approximate $$p(x|y) \approx \int p(x|\theta) q(\theta|y)d\theta$$ with equality if $q=p(\theta|y)$ (i.e. the targeted posterior). Instead of e.g. learning p(x|y) from (x,y). In the case where one only trains $T_x$ we have $q \approx p(\theta \mid x=y)$  which can be a rather bad approximation so $q(x|y)$ will be quite different from $p(x|y)$ and thus even in the case of conditional independence only training $T_x$ would not work well (as somewhat suggested by the experiments). Can the authors clarify if this reasoning is correct, or if I am missing something?

**Limitations:**

The limitations are only partially discussed. In my view, the paper would benefit from a clearer discussion of when the proposed correction strategy may fail. For example:

- when there is significant support mismatch between real observations and simulated data; or when the dimensionality of the observation space makes learning ​ especially difficult.
- **Runtime considerations and ODE integration.** The proposed approach relies on flow matching models that require solving ODEs during training (to generate samples via the learned flows) and during inference. In particular, the training procedure repeatedly integrates the observation-space flow to generate inside the training loop. This introduces a non-negligible computational overhead compared to baselines such as e.g. MF-NPE.

**Strengths And Weaknesses:**

## Strengths
- The paper is well written overall and addresses an interesting and practically relevant problem in SBI.
- The proposed framework is methodologically sound. In particular, the decomposition into an observation-space transport map and a parameter-space correction map is well motivated and appears to work well empirically, especially when both components are trained jointly.
- Good empirical breadth: The experiments span a range of tasks + data/parameter modalities and ablations on the components have been done i.e. Appendix Fig 8. All relevant baselines I am aware off have been ran.

## Weaknesses
- Some of the conceptual claims in the manuscript remain vague or insufficiently supported by the empirical evidence. In particular, the discussion of the setting in which conditional independence between $\theta$ and $y$ would make learning only $u_X$ sufficient is not fully convincing, especially because the ablations seem to suggest that learning only $u_X$ does not perform well in practice. This discrepancy should be explained more clearly (see questions).
- Section 3 is harder to follow than necessary. The exposition is somewhat indirect, with repeated discussion of what one _could_ do before it becomes fully clear what is _actually_ done in the proposed method. A more direct presentation of the final training and inference procedure would improve readability. More generally, the paper would benefit from a sharper separation between intuition, special-case discussion, and the actual algorithm used in experiments.

---

> ### Author Rebuttal · Authors · 2026-03-29
>
> We thank the reviewer for his/her careful reading, interesting questions and remarks.
>
> > [`INDEP`] Role and validity of the Conditional Independence Assumption
>
> Concerns were raised regarding the role of the conditional independence assumption $y \perp \theta | x$ (Eq. 2) in our method. We apologize for the lack of clarity and provide the following clarification, which we will incorporate into the revised paper.
>
> Eq. (2) and (3) are introduced exclusively in the context of related work. They underpin the design of RoPE (Wehenkel et al., 2025), not FMCPE. Our method does not rely on conditional independence at any point. Section 3.2 uses Eq. (2) only to discuss a degenerate special case: if conditional independence held and $\hat{p}_{\theta \mid X}$ were perfect, then learning $T_X$ alone would theoretically suffice to recover $p\_{\theta | Y}$. This is presented as a motivation for the source distribution construction, not as an assumption of our approach. We will rewrite this part to make the distinction explicit and avoid the impression that Eq. (2) is a working assumption of FMCPE.
>
> Regarding the observation that the $T_X$-only ablation study indicates inconsistencies with the above claim, we disagree. The conditional independence does not hold in any of our four experimental tasks and even a perfect $T_X$ cannot fully correct the posterior, since the conditional independence required by Eq. (3) is simply not satisfied. This explains the consistently poor performance of $T_X$-only across all tasks and calibration sizes.
>
> We will add this reasoning to Section 3.2 and to the ablation discussion in Appendix B.
>
> > Section 3 hard to follow
>
> We agree with this assessment. Section 3 was written in a motivational style that introduces several alternatives before converging on the final method, which obscures the actual algorithm. In the revised paper, we will restructure Section 3 by presenting the final training and inference procedure upfront, clearly delimited from the motivating discussion.
>
> > **Q1:** Conditional independence in experimental tasks
>
> In all of our tasks, the conditional independence is not verified. For the Wind Tunnel and Light Tunnel tasks, we redirect the reviewer to (Gamella et al. 2025) for more details about the causal link between the variables at stake. We will add a short sentence in the revised manuscript to clarify this fact.
>
> > **Q2:** About the training of $q_{X|Y}$
>
> In a setting where the conditional independence would be verified, the reviewer is right, since $$ q_{X|Y}(x|y) = \int p_{X|\Theta}(x|\theta)p_{\Theta|Y}\mathrm{d}\theta = p_{X|Y}(x|y)$$
>
> More specifically, the $T_X$-only approach relies on evaluating $\hat{p}_{\theta|X}(\theta \mid \tilde{x})$ at $\tilde{x} = T_X(y)$, which is only theoretically grounded when conditional independence holds. In that case, $\int p(\theta|x)p(x|y)\,dx = p(\theta|y)$, so correcting in the $X$-space alone suffices. When this assumption is violated, as in all our experimental tasks, $T_X$-only lacks theoretical justification and its poor empirical performance is expected.
>
> This further motivates the joint learning of T_theta and T_X in FMCPE, where T_theta can absorb errors that T_X alone cannot correct.

---

> > ### Author Rebuttal · Reviewer_uvU7 · 2026-04-02
> >
> > I thank the clarifications from the authors this resolved my main concern i.e. given that all tasks are not conditionally independent the experimental result on "T_X only" do make sense.
> >
> > I think Q2 is also resolved: Practically one use $(\theta, y)$ i.e. a calibration tuple to get a single true posterior samples from $\theta ~ p(\theta|y)$ which at least asymptotically makes $q(x|y) \rightarrow p(x|y)$, hence T_x will aim to approximate p(x|y) which is sufficient under conditional independence. Although, given that all task do not satisfy conditional independence there are not result showing that just learning T_x works well in this case.

---

> > > ### Author Response · Authors · 2026-04-07
> > >
> > > We thank the reviewer for his/her support, and we plan to add experiments that satisfy the conditional independence assumption (CI) to gain better insights on this case and to complement the experimental validation. As an illustration, we provide results on a toy Gaussian example where we control to which degree the CI holds with a scalar $\alpha$
> > >
> > >
> > >   $$\theta \sim \mathcal{U}([-3,3]^5), \quad \theta \in \mathbb{R}^5 $$
> > >
> > >   $$x \mid \theta \sim \mathcal{N}(A\theta , \sigma_x^2 I), \quad x \in \mathbb{R}^{20}, A \in \mathbb{R}^{20 \times 5} $$
> > >
> > >   $$y \mid \theta, x \sim \mathcal{N}(Cx + \alpha D\theta , \sigma_y^2 I), \quad C \in \mathbb{R}^{20 \times 20}, D \in \mathbb{R}^{20 \times 5}$$
> > >
> > > where $\sigma_x = 0.5$, $\sigma_y = 0.3$.
> > >
> > > When $\alpha = 0$, $y$ depends on $\theta$ only through $x$ (CI holds); when $\alpha > 0$, the term $\alpha D\theta$ introduces a direct $\theta \to y$ effect. We provide metrics values for FMCPE using both $T_{\Theta}$ and $T_X$  and FMCPE using only $T_X$ in the table below. We can clearly see that the gap between both variants is small for $\alpha=0$, suggesting the approach using only $T_X$ can be sufficient. The gap widens as $\alpha$ increases, confirming the positive impact of the $\theta$-flow correction when conditional independence is violated.
> > >
> > >
> > > **Table : Flow-$\theta$ ablation, $N_{\text{cal}}$ = 1000. Mean ± std over 3 seeds. The gap between FMCPE (full) and FMCPE ($T_X$ only) is reported relatively to the value of FMCPE (full) for each line.**
> > >
> > > | $\alpha$   | Method          | J-C2ST ($\downarrow$ 0.5)  | $\Delta$%    | J-MMD ($\downarrow$ 0)       | $\Delta$%|
> > > |-|-|-|-|-|-|
> > > | 0.0 | FMCPE (full)    | .576 ± .006      |       | .00040 ± .00001    |        |
> > > |     | FMCPE ($T_X$ only)  | .613 ± .003      | +6.4% | .00043 ± .00001    | +7.5%  |
> > > | 0.5 | FMCPE (full)    | .630 ± .015      |       | .00045 ± .00000    |        |
> > > |     | FMCPE ($T_X$ only)  | .673 ± .004      | +6.8% | .00051 ± .00001    | +13.3% |
> > > | 1.0 | FMCPE (full)    | .691 ± .016      |       | .00030 ± .00003    |        |
> > > |     | FMCPE ($T_X$ only)  | .777 ± .003      | +12.4%| .00048 ± .00001    | +60.0% |
> > > | 2.0 | FMCPE (full)    | .768 ± .041      |       | .00022 ± .00003    |        |
> > > |     | FMCPE ($T_X$ only)  | .864 ± .004      | +12.5%| .00045 ± .00001    | +104.5%|
> > > | 5.0 | FMCPE (full)    | .545 ± .051      |       | .00033 ± .00017    |        |
> > > |     | FMCPE ($T_X$ only)  | .709 ± .168      | +30.1%| .00052 ± .00008    | +57.6% |

---

### Official Review · Reviewer_uSAM · 2026-03-12

**Soundness:** 3
**Presentation:** 4
**Significance:** 3
**Originality:** 3
**Overall Recommendation:** 5
**Confidence:** 3

**Summary:**

The paper proposes a method for simulation-based inference (SBI) with robustness to misspecification. A joint flow matching algorithm is used to learn two flows between (i) an SBI approximate posterior and high-fidelity parameters samples ($\theta_0$ to $\theta_1$) and (ii) noisy data distribution and approximate simulator model ($x_0$ to $x_1$). Each of these flows are conditioned on the actual data $y$ (associated with $\theta_1$).

**Compliance With Llm Reviewing Policy:**

Affirmed.

**Final Justification:**

The authors rebuttal and comments have convinced me that the terminology used is in line with previous literature, and that there are situations in which a small calibration set is available (typically in experimental science). Part of my original issue with the paper is (though well written) some crucial framing and details were left out. The authors commitment to rectifying these issues, as illustrated by there comments has led me to increasing my score.

**Key Questions For Authors:**

1. How exactly is the misspecification framework (eq. 2 and 3) utilised in this work? Or is it just part of the literature review?

2. What (conditional) distribution does your algorithm target? It was not clear to me how to derive this from Algorithms 1, 2 and Equation 10 (and distributional statements that follow at the top of page 5).

3. Relatedly, do these distributional statements (top of page 5) agree with the sampling done in Algorithm 2? It was not clear to me. Can you explain further?

4. What role is the normal distribution assumption for $x_0$ play? Does this imply misspecification has this particular parametric form?

5. What is the role of the toy Gaussian model in the experiments? What aspect of misspecification does it test?

6. Did you discuss the relative computational cost somewhere other than the appendix? It would be good to provide guidance on this.

**Limitations:**

Yes.

**Strengths And Weaknesses:**

Soundness: The work is technically sound in execution and use of tools. The components of the algorithm are justified and used effectively. However, I found it difficult to understand the overall target of the joint flow matching. For instance, if each component were working perfectly (no Monte Carlo error etc) what would the Algorithm 1 converge to?

The experiments are thorough and show the algorithm to be performing well against suitable alternatives. The computational cost of the method also seems to be relatively low compared to competing methods (but this is not completely clear).

Presentation: I found the paper well written with concise but informative explanations of background knowledge. The experiments were explained and discussed convincingly. I wasn't able to fully understand the motivation of the toy Gaussian example.

Significance: Misspecification is an important problem, and it appears the authors have made progress on this for simulation-based inference. It also appears that their method is computationally efficient, though this is not discussed in the main paper.

Originality: The general setup and joint training appears to be original, and potentially lead to follow up work. Understanding the overall target of the algorithm would enable a better evaluation of originality.

---

> ### Author Rebuttal · Authors · 2026-03-28
>
> We thank the reviewer for the positive assessment of our work and answer all questions below.
>
> > **Q1:** How exactly is the misspecification framework utilised in this work?
>
> We do not explicitly use Eq. (2) and (4) in our work. These are only introduced to state that we do not make such assumption in our work, although it motivated some part of it.
>
> > **Q2:** What (conditional) distribution does your algorithm target?
>
> [`CONVERGENCE`] We provide the following clarification about the target distribution of our method, which we will add as a Remark in Section 3.3 of the revised paper. In the limit of infinite calibration data and perfect optimization, the joint training procedure of Algorithm 1 converges to the following targets:
>
> - First, $\hat{u_X}$ converges to the vector field transporting the base distribution $N(x|y, \sigma^2I)$ to $q_{X|Y}(x|y)$, as defined in Eq. (7). This means the observation-space flow $T_X$ then provides a stochastic map from any real observation $y$ to some $x$ in the simulator space distributed as (7).
> - Second, given a converged $T_X$, the source distribution $\pi_{\theta|Y}(\theta|y)$ becomes a well-defined and informative proposal for the true posterior. The parameter-space flow $\hat{u_{\Theta}}$ then converges to the vector field transporting this source toward $p_{\theta|Y}(\theta|y)$ — the true posterior.
> - The target of the full algorithm is therefore $p_{\theta|Y}$, the true posterior distribution over parameters given real observations.
>
> The intuition behind the joint design is that reducing the distributional gap between source and target makes the flow easier to learn from few calibration samples. The joint training further ensures that $\hat{u}_Θ$ is trained to be robust to imperfect samples from the evolving source distribution, which is critical at inference time when the source may generate imperfect candidates for unseen observations.
>
> We acknowledge that a formal convergence analysis with finite-sample guarantees is beyond the scope of the current work and remains an important direction for future research.
>
> > **Q3:** Do these distributional statements agree with the sampling done in Algorithm 2?
>
> We realize we may not have been sufficiently clear regarding the exact sampling procedure used in Algorithm 2. We provide below additional details clarifying the correspondence between the distributional statements at the top of p.5 and the steps of Algorithm 2.
>
> - **1:  sampling from $p_{\Theta,Y}$:** The statement $(\theta_1, y) \sim p_{\Theta,Y}$ means we sample from the joint distribution, which in practice corresponds to drawing from the calibration set $\mathcal{D}_{\text{cal}}$. This is exactly what is done in this step of the algorithm.
>
> - **2: sampling from $q_{X|Y}$:** This step amounts to sampling from the mixture
> $$q_{X|Y}(x|y) = \int p_{X|\Theta}(x|\theta) p_{\Theta|Y}(\theta|y) d\theta.$$
> To do so, one first draws $\theta_1 \sim p_{\Theta|Y}(\cdot|y)$ (obtained from Step 1), and then samples from the simulator likelihood $p_{X|\Theta}$, i.e. $x_1 \sim S(\theta_1, \varepsilon)$, where $\varepsilon$ denotes the simulator's internal noise. The resulting $x_1$ is an exact sample from $q_{X|Y}$ as defined in Eq. (7); no approximation is introduced beyond the implicit definition of the likelihood via the simulator.
>
> - **3:  sampling from ${N}(x_0;\, y,\, \sigma^2 I_d)$:** We use the observation $y$ drawn from the calibration set and sample $x_0$ accordingly, as stated in the distributional statement on p.5.
>
> - **4–6: sampling from $T_X$:** These steps correspond to sampling from the current approximation $T_X$ of $q_{X|Y}$ at the current training epoch.
>
> - **7:  sampling from $\pi_{\Theta|Y}(\theta_0|y)$:** Once convergence is reached, $\tilde{x} = T_X(x_0, y)$ approximates a sample from $q_{X|Y}$, and sampling $\theta_0 \sim \hat{p}_{\Theta|X}(\theta|\tilde{x})$ therefore corresponds to sampling from the mixture defined in Equation (6)
>
> > **Q4:** Role of the normal assumption for $x_0$
>
> The Gaussian source for $x_0$ is an arbitrary but empirically convenient choice motivated by Albergo et al. (2024). It could be replaced by any other distribution without implying any assumption on the nature of the misspecification.
>
> > **Q5:** Role of the toy Gaussian model in the experiments?
>
> The Gaussian task (Eq. 11) is a controlled benchmark where the true posterior is available in closed form, allowing exact posterior-level metrics rather than joint-distribution proxies. Despite its simplicity, it already reveals failure modes for several benchmarked methods.
>
> > **Q6:** About the computational cost
>
> We discussed the method computational cost at the end of Section 4.3, but we plan to extend this discussion to provide insights on which parameters can be tuned to reduce costs, e.g. number of ODE steps.
>
> Reference
> - Stochastic interpolants with data-dependent couplings, Albergo et al., 2024

---

> > ### Author Rebuttal · Reviewer_uSAM · 2026-04-02
> >
> > **Q1**:
> > I think that the use of ‘model misspecification’ is blurry here, perhaps even misused. The assumed model (for the SBI component) is a computationally efficient means to construct a good amortised approximate posterior distribution as a base distribution for flow matching. The true model is not specified analytically but rather represented via the small calibration data set (which includes true data-parameter pairs). Methods addressing model misspecification in the usual sense only assume access to data from the model (no corresponding true parameters). As such, I suggest that the paper is closer to statistical emulation than SBI for misspecified models. Personally, I was not expecting a method for ‘SBI under model misspecification’ to assume access to samples from the joint distribution over data and parameters. If my understanding is correct, I would suggest that even the paper title is misleading: ‘Approximate SBI for efficient flow matching’ or similar would be more appropriate.
> >
> > I can see how methods for dealing with model misspecification have informed the SBI approach (to be fed into flow matching). But I have lingering substantial concerns about the framing of the paper.
> >
> > [Reviewer Qmdc](https://openreview.net/forum?id=Ke4poDVYTO&noteId=KFoVHiqK1a) raises a similar point which, in my opinion, is not sufficiently addressed by your rebuttal. True parameter-data pairs are rarely ever available for machine learning or statistical inference tasks, so your method is only narrowly applicable (to emulation for example).
> >
> > **Q2**-**Q6**: The further information provided and promised updates to the manuscript resolve these.
> >
> > **Summary**
> >
> > In light of this acknowledgement I have reduced my score of the paper to 2, with higher uncertainty. I encourage the author's to respond, if they think my understanding is incorrect. It is possible that the term 'model misspecification' is used more broadly in an area of ML I am unfamiliar with, for example. It is also possible that there are many more settings (beyond emulation) where true data-parameter pairs (at least in small numbers) are available. But at least two other reviewers share these concerns, so I think it is unlikely.
> >
> > **Update after [author reply](https://openreview.net/forum?id=Ke4poDVYTO&noteId=TbDpiiKvxj)**
> >
> > You are correct that a string of recent literature has used a small calibration set to correct a misspecified simulator in SBI (Wehenkel et al.,  2025; Senouf et al., 2025; Krouglova et al., 2025). Thank you for pointing this out here again. I would still argue that it is not a common approach to misspecification in SBI. The recent review (Kelly et. al., 2025) contains several methods for addressing model misspecification in SBI, without assuming a small calibration set is available. I would argue that no access to a calibration set is the more common situation and this should be better addressed in the paper. This along with a very clear scoping of the paper and examples would provide much needed clarity.
> >
> > I have updated my score accordingly. Thank you for providing further information.

---

> > > ### Author Response · Authors · 2026-04-02
> > >
> > > We thank Reviewer `uSAM` for the continued engagement and for explicitly identifying the points where clarification may resolve the disagreement. We address them directly, and also take the opportunity to clarify the related concern raised by Reviewer `Qmdc`.
> > >
> > > We also note that, upon reflection, our initial response to Q1 focused on clarifying the algorithm's target distribution rather than addressing the broader framing concern — we apologize for this misunderstanding and address it directly here.
> > >
> > > **Our setting of misspecification in SBI follows RoPE (Wehenkel et al., ICML 2025), inductive RoPE (Senouf et al., NeurIPS 2025), and MFNPE (Krouglova et al., 2025)**
> > >
> > > The reviewer correctly identifies that their concern may stem from a difference in how this term is used in SBI versus classical statistics — and Reviewer Qmdc raises the same point, noting that both Walker (2013) and Frazier et al. (2020) are cited without a clear definition being given upfront. We acknowledge this as a genuine presentation gap and clarify here.
> > >
> > > In classical settings such as Walker (2013), misspecification refers typically to maximum likelihood estimation under a wrong parametric model, with no simulator and no parameter-data pairs available. This is not the setting we address, and our citation of Walker (2013) was only intended to provide broader context — we should have made this explicit.
> > >
> > > In the SBI literature, misspecification refers specifically to the mismatch between the simulator's likelihood $p_{X|\theta}$ and the true data-generating likelihood $p_{Y|\theta}$ (Frazier et al., 2020; Cannon et al., 2022): the simulator may be inaccurate due to mathematical approximations or simplified dynamics, yet remains useful as a cheap approximation for training a posterior estimator. Our setting is then: given this misspecified simulator and a small set of calibration pairs $(\theta, y)$ from the true joint — coming from controlled experiments or a higher-fidelity model — correct the simulation-based posterior toward the true posterior. This is precisely the formulation used by the most directly related prior works in this space, including RoPE (Wehenkel et al., ICML 2025), inductive RoPE (Senouf et al., NeurIPS 2025), and MFNPE (Krouglova et al., 2025):
> > >
> > > - **RoPE** and **inductive RoPE** : assumes ${(\theta_i, y_i)}$ calibration pairs, frames the problem as SBI under misspecification.
> > > - **MFNPE** : uses high-fidelity $(\theta, y)$ pairs to refine a misspecified posterior, same framing.
> > >
> > > We will add an explicit definition of misspecification in the SBI sense to the introduction to prevent this confusion in the revised manuscript.
> > >
> > > **On the availability of $(\theta, y)$ pairs beyond emulation**
> > >
> > > Reviewer `uSAM` asks whether such pairs are available in settings beyond emulation. They are, and they represent the core application domain of SBI — experimental science — where parameters are frequently controlled inputs that the experimenter sets and records directly. In addition they are of more and more critical importance to sustain the development of efficient and accurate digital twins in high societal impact domains:
> > >
> > > - Gravitational wave inference (Dax et al., 2021): injected signals with known parameters into real detector noise — $(\theta, y)$ pairs by construction, standard LIGO practice.
> > > - Remote sensing: for forest surveillance, drones provide observations interpreted through inherently imperfect models. Experts can collect calibrated field data, which can be challenging in dense, remote regions such as the Amazon. Similarly, monitoring snow caps is economically important when using hydropower; to offset limits in models, experts install ground instruments on high-altitude glaciers.
> > > - Computational neuroscience: biophysical neuron models calibrated against patch-clamp recordings where the stimulus parameters are precisely controlled by the experimenter.
> > > - WindTunnel / LightTunnel (our experiments): hatch angle and polarizer angle are physically controlled and logged — calibration pairs arise from standard controlled experiments.
> > >
> > >
> > >
> > > In all these cases, N_cal = 10–50 calibration pairs require only controlled experiments. The intuition that ground-truth parameters are "rarely available" holds in pure ML settings (e.g., image classification) but does not transfer to the experimental sciences SBI is designed for.
> > >
> > > **On the title and framing**
> > >
> > > We are committed to updating the title to emphasize calibration (e.g., "Flow Matching Calibration for SBI under Model Misspecification") and to adding an explicit introductory paragraph drawing the distinction between Walker-style misspecification and the SBI notion we address. This should directly resolve the concern shared by both Reviewer `uSAM` and Reviewer `Qmdc` that the scope was not immediately apparent.
> > >
> > > We hope this clarifies our positioning and respectfully invite Reviewer uSAM to reconsider their score.

---

### Official Review · Reviewer_xKrR · 2026-03-12

**Soundness:** 3
**Presentation:** 3
**Significance:** 3
**Originality:** 3
**Overall Recommendation:** 4
**Confidence:** 3

**Summary:**

This paper studies simulation-based inference (SBI) under model misspecification and proposes a two-stage correction method based on flow matching. The approach first learns an observation-space transport that maps a real observation toward simulator-compatible surrogates, then uses these surrogates to construct a simulator-informed source distribution for a second flow in parameter space that aims to correct the posterior. The method is designed to leverage information from an imperfect simulator together with a small number of calibration pairs from the real or high-fidelity process.

**Compliance With Llm Reviewing Policy:**

Affirmed.

**Key Questions For Authors:**

1. Could the authors clarify more explicitly in the main text the distinction between ideal target distributions and the finite-sample approximations actually used in training? In particular, the transition from $q_{X|Y}$ as defined in Equation (7) to the empirical procedure based on calibration pairs and simulator draws deserves more clarification.

2. The source for $u_\Theta$ depends on the evolving $u_X$, making the effective source distribution non-stationary during training. Have the authors explored more systematically how sensitive this is to optimizer choice, learning rate, gradient clipping, or the stop-gradient operation in Algorithm 2?

3. The same limited calibration pairs supervise both the x-space and $\theta$-space components. Have the authors considered split-calibration or cross-fitting variants to assess whether the observed gains persist when the two learned flows do not reuse exactly the same scarce pairs?

4. For most tasks, evaluation relies on joint-distribution metrics such as jC2ST and joint Wasserstein because only one ground-truth $\theta_j$ is available per $y_j$. Do the authors have additional posterior-calibration diagnostics, such as empirical coverage or simulation-based calibration style checks, that could support the calibration claims more directly?

**Limitations:**

My main reservation is that the theoretical framing is stronger than what is actually established. The key quantity $q_{X|Y}$ is defined using the unknown true posterior $p_{\Theta|Y}$, i.e. the very object the method is trying to estimate. In practice, the paper replaces direct access to this object with sparse calibration pairs $(\theta_j,y_j)$ and simulator samples $x_j \sim p_{X|\Theta}(\cdot\mid \theta_j)$. That is a reasonable way to approximate the target from samples, but it leaves open the harder question of whether the method can generalize this correction reliably from a finite and potentially very small calibration set.

A second limitation is that the same scarce calibration pairs are reused across tightly coupled components. The pair $(\theta_1,y)$ is used both to generate simulator targets $x_1$ for the observation-space flow and to provide target parameters for the parameter-space flow. This is efficient, but it also means the method leans heavily on the same limited supervision in more than one place.

A third limitation is that the evaluation metrics for most tasks are indirect with respect to posterior calibration. Because only one ground-truth parameter is available per observation outside the Gaussian task, the paper relies on jC2ST, joint Wasserstein distance, MMD, and MSE computed on joint samples $(\tilde\theta_j,y_j)$. These are useful discrepancy measures, but they do not directly validate the conditional posterior or its calibration in a Bayesian sense.

**Strengths And Weaknesses:**

The paper addresses a fundamental problem in simulation-based inference: how to correct a simulator-trained posterior when the simulator is misspecified and only a small number of real or high-fidelity calibration pairs is available. This is an important problem, particularly in scientific settings where SBI is attractive because simulators are available, but imperfect. In that sense, the paper has clear practical relevance. The impact might be more specialized than broad, but that is appropriate for the contribution. My main reservation is that the practical significance ultimately depends on whether the method remains stable and sample-efficient beyond the settings evaluated here. So I view the significance as real and meaningful, but more as a promising direction than as a clearly established shift in practice.

In terms of originality, the paper is reasonably novel. The contribution is not flow matching alone, nor simulator-based posterior estimation alone, but the specific way these ingredients are combined: first learning an observation-space transport that maps real observations toward simulator-compatible surrogates, then using those surrogates to define a simulator-informed source distribution for a second flow in parameter space. That is a nontrivial and well-motivated synthesis, and the distinctions from RoPE and MFNPE are meaningful.
The strongest aspect of the paper is the simulator-informed source construction in Section 3.2. Rather than feeding a real observation directly into a simulator-trained posterior, the method first learns a transport into simulator space and then uses the simulator posterior only as a proposal for a second correction step. This is conceptually clever and well matched to the misspecification problem the paper is trying to solve.

A second strength is that the paper is not purely conceptual. It provides concrete algorithms, explicitly discusses training stability, and evaluates the method against relevant baselines on both synthetic and real-world tasks. Even if I have some reservations about how strongly the empirical section supports the main claims, the experimental setup is more substantial than a minimal proof of concept.
The paper is reasonably well structured and the high-level narrative is easy to follow. The motivation is clear, the progression from background, theory to method is logical.

The main weakness of the method is that the paper sometimes presents the construction as more direct and more general than it really is.
In practice, the model does not have access to the true posterior. It only has sparse calibration pairs $(\theta, y)$ from the joint distribution and must reconstruct the conditional posterior from that limited signal. That is a much harder problem than the presentation sometimes suggests. Whether the approach succeeds depends heavily on how many calibration pairs are available, how well they cover the relevant observation space, how complex the posterior family is, and how informative the simulator-based source already is. Samples from the true joint process can support learning the true posterior, but they do not provide it directly, and I think the paper understates that gap.

The manuscript seems to overstate the generality of the approach. The introduction claims that the method (i.e. flows) is “agnostic to the type of misspecifications” because it does not require an explicit parametric model of the mismatch. That is true only in a limited sense. The method may avoid specifying an analytic error model, but it still assumes access to calibration pairs from the real or high-fidelity process and assumes that these sparse pairs are informative enough to learn a meaningful correction which goes back to my point before. That is already a fairly strong structural assumption. The paper also tends to blur two separate claims: that the method can correct misspecification in principle, and that it can do so efficiently with very small calibration sets. Those are not the same, and the latter does not automatically follow from the former.
A further weakness is that the full construction stacks several learned approximations on top of one another. The observation-space flow, the simulator-informed source distribution, and the final parameter-space correction are all tightly coupled and all depend on the same limited calibration data. This makes the method elegant, but also potentially fragile: if one stage is poor, the later stages must absorb that error, and it becomes difficult to diagnose where failure originates. The method is technically interesting, but I think the paper understates how strongly its success depends on finite-sample coverage, source quality, and the severity of misspecification.

Overall, the paper is interesting and methodologically thoughtful. The core idea is original enough to matter, and the problem it addresses is important. My main hesitation is that the theoretical story is more heuristic than rigorous, and the empirical validation, while useful, remains somewhat indirect with respect to the strongest claims about posterior correction and calibration.

---

> ### Author Rebuttal · Authors · 2026-03-28
>
> We thank the reviewer for the positive feedback on the significance and originality of our work.
>
> > The main weakness of the method is that the paper sometimes presents the construction as more direct and more general than it really is
>
> We agree that some parts overstate the directness of the construction and will revise the paper to more clearly distinguish the idealized formulation, where $q_{X|Y}$ and $p_{\theta|Y}$ are treated as known targets, and the practical implementation, where both must be approximated from sparse calibration pairs. In particular, we will add an explicit paragraph after Eq. (7) acknowledging that:
> - $q_{X|Y}(x|y)$ is never directly observed but approximated via simulator draws conditioned on calibration parameters
> - the quality of this approximation depends on how well the calibration set covers the observation space
> - the method's success with small N_cal is an empirical finding, not a theoretical guarantee.
>
> We believe that making these distinctions explicit will strengthen rather than weaken the paper, by giving practitioners a clearer picture of where the method is likely to work well and where it may struggle.
>
> > About our method being agnostic to misspecification
>
> To be precise, FMCPE is agnostic to the parametric form of misspecification, unlike e.g. Ward et al. (2022). However, it does require access to a small calibration set $D_{cal}$ drawn from $p_{Θ,Y}$, and its efficiency depends on calibration set coverage, posterior complexity, and misspecification severity. We will make these dependencies explicit in the introduction and limitations section, and correct "applicable as post-processing to any SBI model" to "applicable as post-processing to any SBI model, given access to a small calibration set."
>
> > The paper tends to blur two separate claims: that the method can correct misspecification in principle, and that it can do so efficiently with very small calibration sets.
>
> See [`CONVERGENCE`] in our answer to reviewer uSAM.
>
> > **Q1:** Transition from $q_{X|Y}$ as defined in Equation (7) to the empirical procedure
>
> We will adjust the manuscript to make the link between the target $q_{X|Y}$ and the sampling procedure clearer.  Indeed, the ideal target in Eq. (7) is:
> $$q_{X|Y}(x|y) = \int p_{X|\theta}(x|\theta) p_{\theta|Y}(\theta|y) d\theta$$ which is doubly intractable — neither $p_{\theta|Y}$ nor the integral are accessible directly. In practice, we approximate this as follows: for each calibration pair $(\theta_j, y_j) \sim p_{\theta,Y}$, the $\theta_j$ is already a sample from $p_{\theta|Y}(\cdot|y_j)$ by definition, so drawing $x_j \sim p_{X|\theta}(\cdot|\theta_j)$ via the simulator yields a sample $x_j \sim q_{X|Y}(\cdot|y_j)$. This is an unbiased single-sample Monte Carlo estimate of $q_{X|Y}(\cdot|y_j)$ per calibration observation. The finite-sample approximation error therefore has two sources: (a) the number of calibration pairs N_cal, which controls how well the marginal distribution of $y$ is covered; and (b) the single simulator draw per calibration pair, which could in principle be replaced by multiple draws per $(\theta_j, y_j)$ to reduce variance at the cost of more simulator calls. We will make this decomposition explicit in the text, and acknowledge that the quality of the approximation degrades when the calibration set is small or poorly covers the observation space of interest.
>
> > **Q2:** Sensitivity to optimizer / learning rate / gradient clipping
>
> We conducted convergence experiments across a range of hyperparameter configurations and did not observe any critical failure modes within the ranges we explored. As a concrete illustration, we provide a table [here](https://osf.io/tkwqv?view_only=b57cb44410324d1dbd111aadd604551b) for the pendulum case in which we systematically varied the learning rate and gradient clipping threshold.
> Regarding the stop-gradient operation in Algorithm 2, this is a deliberate design choice rather than a tunable hyperparameter: propagating gradients through the ODE sampling procedure would be computationally prohibitive, and we therefore do not consider it as an alternative.
>
> > **Q3:** Split-calibration / cross-fitting
>
> This is a legitimate concern. Split-calibration and cross-fitting could yield substantial improvements when sufficient calibration data are available. However, in our setting, empirical finding suggest that the limited amount of calibration data means that splitting it would likely hurt performance more than it would help. We nonetheless view this as a promising direction for future work.
>
> > **Q4:**  Posterior calibration diagnostics (coverage, SBC)
>
> To strengthen our empirical claims, we will include SBC and coverage diagnostics in the revised version of the paper. In the meantime, we provide results for these diagnostics on the Gaussian and Wind Tunnel tasks [in this link](https://osf.io/m785j/overview?view_only=b57cb44410324d1dbd111aadd604551b).

---

> > ### Author Rebuttal · Reviewer_xKrR · 2026-04-02
> >
> > Thank you for the detailed rebuttal. The response addresses my concerns in a meaningful way. I especially appreciate the clearer distinction between the idealized targets and the empirical finite-sample procedure, as well as the more careful wording around what "agnostic to misspecification" means in practice. The added coverage and SBC diagnostics are also a meaningful improvement and provide more direct support for the calibration claims. They show clear gains over uncorrected NPE, though not universal dominance over all baselines.
> >
> > My reservation is only partially resolved: the exposition is now much better calibrated, but the finite-sample theoretical story remains limited, and the split-calibration / reuse-of-calibration-data concern is still largely open. Overall, the rebuttal increases my confidence in the work, but I still view some of the strongest claims as more heuristic than fully established.

---

> > > ### Author Response · Authors · 2026-04-07
> > >
> > > We thank the reviewer for his/her acknowledgment and for pointing out the need for further clarifications.
> > >
> > > **-- follow-up to Q1: the finite-sample theoretical story remains limited**
> > >
> > >
> > > We agree that a finite-sample theoretical analysis would strengthen the paper. We view this as important directions for future work, which are still largely open questions in the broader SBI-under-misspecification literature as well — to our knowledge, none of the existing methods we compare against (RoPE, MFNPE) provide finite-sample guarantees either.
> > >
> > >
> > >
> > > **-- follow-up to Q3: split-calibration / reuse-of-calibration-data concern**
> > >
> > > This point is very interesting and we agree that it would require further theoretical thoughts.
> > >
> > > On the experimental view point, we made 2 new sets of experiments for the Gaussian and Wind tunnel tasks, splitting the calibration set in 2 and using then different pairs to learn the X-flow and $\theta$-flow.
> > > For these tasks and up to $N_{cal}=1000$ we see a decrease in performance (see results [here](https://osf.io/m785j/files/4e5hg?view_only=b57cb44410324d1dbd111aadd604551b)) which is probably due to the reduction by 2 of the calibration set size.
> > >
> > > Regarding the strongest claims being "more heuristic than fully established": we take this point seriously and will revise the paper to more carefully distinguish between what is empirically demonstrated and what remains conjectural. In particular, we will temper claims about small-$N_\text{cal}$ efficiency to present them as empirical findings rather than established properties of the method.

---

### Official Review · Reviewer_Qmdc · 2026-03-15

**Soundness:** 3
**Presentation:** 2
**Significance:** 3
**Originality:** 3
**Overall Recommendation:** 3
**Confidence:** 4

**Summary:**

The paper addresses the challenge of simulation-based inference using flow matching calibration; with a particular focus on model misspecification. The assumption is that abundant simulation data is available for training a posterior estimator; and then a limited number of calibration samples from real observed data are also available. Comparisons between the proposed approach and three other approaches are made on four benchmark problems with up to four parameters to be estimated.

**Compliance With Llm Reviewing Policy:**

Affirmed.

**Final Justification:**

Author discussion has clarified some situations when the method is likely to work well, provided statistical grounding, and provided additional empirical evidence that calibration works well in the provided experiments.

The author rebuttal addressed my main concerns about:
- The confusion over the problem being solved (defining model mis-specification) and similarity to other methods assuming there are $(\theta, y)$ calibration pairs. This concern was primarily addressed in the discussion with reviewer uSAM.
- Additional empirical plots that I requested showing that calibration moves the mode of the distribution close to the known ground-truth parameter (not just narrowing a distribution).

Authors also added mathematical definitions for the problem being solved and tied this to the steps in the algorithm. I do not think that there are any theoretical claims about flow-matching provided. Also, empirical analysis still only has ground-truth in the Gaussian toy example; so it is difficult to make strong, general claims.

**Key Questions For Authors:**

The paper claims this method is agnostic to the form of model mis-specification. It also states that the simulator is assumed to have "random noise". That assumption matches the assumption of Equation (2). But then the paper also says that the assumption of Equation 2 is not an issue. There is an inherent tradeoff to being agnostic, and so we would expect that other methods will perform better than this approach when the task matches their assumptions. Could you discuss this tradeoff and clarify whether there are any assumptions about model mis-specification that your approach works under?

Could you give a comparison of how all of these methods work when there is no model mis-specification? Could that be used as a ground truth posterior estimate?

Could you show the posterior estimates from the proposed approach before and after calibration? This would help illustrate how model mis-specification is affecting the standard SBI approach and how calibration fixes it. Some of the approaches get broader (more uniform) posterior distributions with more calibration data, which may indicate that calibration is inducing a mixture of the simulation-based posterior and the calibration-based posterior.

**Limitations:**

Yes

**Strengths And Weaknesses:**

Soundness


[Authors have addressed this concern in the discussion and will update the paper to carefully define model misspecification and clarify how it is related to that in RoPE, MFNPE]. Need to define model misspecification. Model misspecification means something different in simulation-based inference (Frazier 2020 and Schmitt 2024) versus in Bayesian inference (Walker 2013). Both definitions are cited in the paper, but it is seems to be the simulation model that is mis-specified (not the distribution model). Discussion of how the proposed approach relates to existing approaches would be made much more clear with a clear definition of model misspecification.

Is there theoretical justification for flow matching to perform well under model misspecification? A clear definition of model mis-specification and discussion of the mathematics used in related approaches would help motivate the new approach. The arguments given suggest the “imperfections” will be reduced. Defining these imperfections would clarify how they can be reduced. Flow matching can fail catastrophically, so it is important to understand under what circumstances the proposed approach is likely to work well.

“applicable as a post-processing to any SBI model.” Is not true if there is no calibration data available.

[Additional empirical results are provided that show that calibration moves the mass of the distribution in better alignment with the known parameter. We still do not know if "tight around the ground truth" is the right distribution, but at least we can see that the major impact that calibration has is significantly moving the mode of the distribution (not just sharpening).] The biggest problem with interpreting the results is that there is no ground-truth posterior to compare against. The paper implies that it is good to have a tight peak around a single result that includes the one ground-truth parameter value that produced the observation. There is no reason to believe that would be true for all tasks.

[Limitations and claims have been discussed and will be incorporated into the paper. Mathematical definitions of the posterior to be calibrated are being added to the paper.] Claims have limited support based on experiments on four tasks. Theoretical analysis would strengthen the claims. Strengths of the approach are evaluated by weaknesses are not discussed.


Presentation

This statement on Page 3 should be explained earlier to put the work into context of existing context. “Our method follows the same spirit of leveraging low-fidelity approximations obtained from a simulator, but uses them differently: the low-fidelity estimator plays a distinct role, and the subsequent corrections it undergoes are of a different nature, as detailed in the next section.”

The conditional independence assumption of Equation (2) seems unlikely to handle bias introduced by the misspecification. Bias is stated as one of the main concerns. Section 3.2 claims that this is assumption is “no longer an issue” but is there any guarantee for the proposed approach? Section 3, first paragraph, states that this paper assumes a stochastic simulator with random noise in the generative process. Yet it also claims that flow matching is used for being “agnostic” to the form of model mis-specification. Re-writing might clarify what the paper is trying to say. Equation (2) is the only specific assumption given in the paper, so I assumed it was relevant to the proposed approach (then later I was told that it is not an issue).

Box-plots: Label y axes. Colors are not visible when variance is small.

Figure 3: The x-axis label on the first row is missing. Is there a reason to put Task B before Task A? Using the actual name in the caption (Gaussian and WindTunnel) would be consistent with the rest of the writing in the paper and lead to less confusion for the reader. The “A” y-axis label on the first row can be confused with calling the task “A” (which is the second row).

Figure 7: A corner plot should be used to show distributions for all of the parameters and all-pairs joint distributions. Also for the 3-parameter Gaussian problem.

Task descriptions: Tasks involve 3, 2, 1, or 4-dimensional parameter estimation. In the results, it is mentioned that the Gaussian posterior should be uni-modal. Is this true of the other tasks as well? Model mis-specification is only described for Pendulum and LightTunnel of the tasks. Stating these things clearly upfront in the task descriptions would clarify why these particular tasks are chosen; including how they demonstrate various strengths/weaknesses of the proposed and compared methods.

Figure 3 top-left plot indicates that this approach can fail (the true value lies in a zero-probability space of the estimated posterior). Yet, there is no discussion about detecting failure of the posterior estimate or insight as to when the method performs well versus what the limitations are.



Significance

The paper addresses an important challenge in simulation-based inference. However, the discussion of results gives little insight that could be used by practitioners. The lack of theoretical motivation for the approach gives little insight into what scenarios the approach is likely to work well, how to detect whether the method fails, or giving insight into why the method solves specific inference challenges with few calibration samples. Author discussion has clarified some situations when the method is likely to work well, provided statistical grounding, and provided additional empirical evidence that calibration works well in the provided experiments.


Originality

The originality lies in using flow matching on observed data to refine a simulation-based posterior estimate. Flow matching is suggested as requiring fewer calibration samples than other approaches.

---

> ### Author Rebuttal · Authors · 2026-03-30
>
> ## Response to reviewer `Qmdc`
>
> Thanks for your detailed analysis and suggestions for improving our presentation.
>
> ### Remarks
>
>
> > A clearer definition of  misspecification:
>
> Indeed, the reference to Walker 2013 creates confusion. As written in l.61 our definition of misspecification relates to the simulator. Model misspecification as used in Walker 2013 or more recently in Tomaselli et al 2025, is a more general setting not necessarily relevant for SBI. Walker 2013 was cited only to acknowledge a broader literature. Simply removing the sentence where it is cited should remove the ambiguity.
>
>
> > No ground truth posteriors to compare against
>
>
> The reviewer raises a valid point regarding the lack of ground-truth posteriors for most tasks. Only the Gaussian task provides an analytical posterior, and for the remaining tasks we rely on joint-distribution metrics (jC2ST, W₂, MMD, MSE).
>
> In revisions, we will include additional cases where ground-truth posteriors are available — either in analytical form or through well-converged MCMC — to enable direct posterior-level comparisons across a broader range of tasks.
>
> To strengthen our current claims, we have performed simulation-based calibration (SBC) and report coverage diagnostics for two tasks. These results are available [here](https://osf.io/m785j/files/osfstorage?view_only=b57cb44410324d1dbd111aadd604551b) and will be incorporated into the revised version of the paper.
>
> > This statement on Page 3 should be explained earlier
>
> Thanks, we can add a similar sentence in the introduction to present the context earlier, that will also help to clarify the type of misspecification we are considering as suggested.
>
>
> >  Role and validity of the Conditional Independence Assumption (Eq. 2)
>
> We apologize for the confusing wording. The conditional independence assumption is not required in our method. See also Q1 response below and  response [INDEP] to Reviewer uvU7.
>
> **Improvements in presentation**
>
> Thanks for your suggestions and pointing out presentation issues. In the final version of the paper, we will add the missing x-axis label on Fig. 3 (first row), renamed "Task A/B" to "Gaussian/Pendulum" throughout for consistency, and corner plots for the Light Tunnel (all 4 parameters) and Gaussian (all 3 dimensions) tasks in the appendix. We will also clarify tasks description as suggested.
>
>
>
> ### Questions
>
>
> > **Q1:**  The paper claims this method is agnostic to the form of model mis-specification.
>
> Equation (2) is indeed not an issue, we only  introduced it to contextualize our work within the existing literature. The "random noise" we mention is a consequence of the inherent stochasticity of the simulator and is not connected to any aspect of model misspecification mentioned afterward. In particular we **do not** assume here that $y$ is a noisy version of the simulator output as in $y = S(\theta) + \epsilon$ with $\theta \sim p(\theta)$.
>
> About our method being agnostic to the form of misspecification, FMCPE is rather agnostic to the parametric form of misspecification, unlike e.g. Ward et al. (2022).
>
> We do not make any specific assumptions about model misspecification, we only assume that the simulator provides meaningful information about the true data generating process. We did not elaborate on this as it is a global statement across the SBI literature regarding model misspecification.
>
>
>
> > **Q2** Behavior in a no misspecification setting
>
> We thank the reviewer for raising this important point. We have added this experiment to the appendix using the Gaussian task with identical simulator and real data-generating process ( [here](https://osf.io/m785j/files/ecbdf?view_only=b57cb44410324d1dbd111aadd604551b)). Results confirm that FMCPE degrades gracefully in this setting, recovering performance comparable to standard NPE, with no spurious corrections when none are needed. At low $N_{\text{cal}}$, we observe greater differences with  NPE, but we point out that we still perform better than other methods across all metrics.
>
>
> > **Q3** Posteriors before and after calibration
>
> This is an interesting point. We have provided examples of the posterior before and after calibration for several tasks ([here](https://osf.io/m785j/files/osfstorage?view_only=b57cb44410324d1dbd111aadd604551b)).
>
> The observation that posteriors can become broader after calibration is indeed something we observe in these visualizations. We see this as a rather encouraging aspect when dealing with statistical inference. As pointed out by Ward et al. (2022): conservative posterior estimates are generally preferable to overconfident ones, as the latter could lead to erroneous scientific conclusions.
>
> References:
> - Robust Simulation Based Inference. Tomaselli et al. 2025
> - Robust Neural Posterior Estimation and Statistical Model Criticism, Ward et al. 2022

---

> > ### Author Rebuttal · Reviewer_Qmdc · 2026-04-04
> >
> > I appreciate the additional results in response to Q2 and Q3. The comparisons of pre-calibration and post-calibration are particularly compelling and strengthen the paper significantly.
> >
> > I am still concerned about the lack of clarity in the writing/presentation of the paper, which is difficult to assess without seeing a fully revised paper. The paper is addressing an important question and upon initially reading the paper, I was quite interested in the approach. However, while writing the initial review, I found it difficult to nail down specifics in the paper (defining the problem being solved, assumptions about misspecification, and quantification of the results). Some of my questions were also raised by other reviewers even after the rebuttals; and so it seems that a re-write of the paper to clarify the confusion is necessary.

---

> > > ### Author Response · Authors · 2026-04-07
> > >
> > > We thank the reviewer for his/her acknowledgement. Regarding the lack of clarity in some parts of the paper, we believe it can be addressed with the following additional explanations. Overall, we feel that the clarifications  below can be easily incoporated in the paper and we are grateful to the reviewers for their analysis and in particular to Qmdc for being very specific and so helpful in suggesting where and how to do that. As suggested by the reviewer, we can improve on the 3 main points below:
> > >
> > > 1) Reformulate the problem being solved to avoid confusion with other scopes and insist more on the positioning of our work, in particular with respect to a recent string of papers (eg. Wehenkel et al., ICML 2025; Senouf et al., Neurips 2025; Krouglova et al., ICLR 2026). We have also commited to this clarification in  our acknowledgment response to uSAM.
> > > 2) Explain better that in contrast to other works, we do not make specific assumptions about the *simulator misspecification*.
> > > 3) Expose and discuss more solutions for an improved quantification of the results.
> > >
> > >
> > >
> > > **1) Problem being solved:**
> > > We will clarifiy the problem formulation in the introduction as follows:
> > > >We address the same problem as in a recent line of works: Wehenkel et al., ICML 2025; Senouf et al., Neurips 2025; Krouglova et al., ICLR 2026. More specifically, we consider the setting where observations $y$ are related to a physical parameter $\theta$ according to some true data generative process (DGP) $p(y|\theta)$, which is unknown.  We assume we have access to a simulator $x=S(\theta,\epsilon)$ defining a conditional probability $p(x|\theta)$ and a small calibration dataset of observation/parameter pairs $(y_i,\theta_i)$ from the true DGP. The simulator is misspecified, in the sense that $p(x|\theta)$ is only an approximation of the true DGP. The goal is to infer $\theta$ given an observation $y$ using the simulator while correcting for its misspecification using the calibration dataset.
> > >
> > >
> > > Additionally, to further clarify the positioning, as suggested by the reviewer, we propose to move the text of line l.121-125 right after the problem formulation: "our method follows the same spirit of leveraging low-fidelity approximations obtained from a simulator, but uses them differently [...]".
> > >
> > >
> > > We will also clarify that, in contrast to standard Bayesian inference, where $\theta$ may not have a physical interpretation, we are interested **in SBI applications where both $\theta$ values and observations are quantities of physical meaning and interest**. Typically, in complex applications, it is often impractical to model every detail of the true DGP $p(y|\theta)$ eventhough input $\theta$ and output $y$ of the phenomenon under study are clearly identified. We propose to add these discussions after l.60-61 where it is stated that the term misspecification is used to refer to a mismatch between the simulation model and the true DGP.
> > >
> > >
> > >
> > >
> > >
> > >
> > > **2) Assumptions about misspecification**
> > > In doing the above, one originality of our work is not to rely on a conditional independence assumption (eq (2)), which may not be true in practice.
> > > Instead (l.135-138), we assume a random noise $\varepsilon$ accounting for randomness in the generative process but not related to the form of model misspecification. In particular, we do not assume that $y$ is a noisy version of $x$ as in $y = S(\theta) + \varepsilon$, which would ensure that eq. (2) holds. We can easily add this latter specific example to illustrate eq. (2) and what it typically implies.
> > >
> > >
> > > **3) Quantification of the results**
> > > Regarding this concern, if it refers to the lack of ground-truth posteriors, we can easily insist in the paper that this is a common setting in related work (Wehenkel et al., 2025; Krouglova et al., 2026), where joint-distribution metrics are standard practice. That said, we agree that additional diagnostics strengthen the paper. The simulation-based calibration (SBC) and coverage results we have produced for two tasks (available [here](https://osf.io/m785j/overview?view_only=b57cb44410324d1dbd111aadd604551b)) can be straightforwardly incorporated into the appendix of the revised version.

---

### Decision · Program_Chairs · 2026-04-30

**Decision:**

Accept (regular)

**Comment:**

This paper is on the subject of simulation-based inference (SBI), and in particular seeks to address the challenge of model misspecification. The paper proposes a flow-matching corrected posterior estimation framework that refines SBI-trained estimators using a small calibration set.

Reviewers have generally had a positive view on this paper, and they especially appreciated the empirical depth of the paper. In particular, the rebuttal helped clarify many reviewer concerns. During the discussion, several points came up as the main concerns left to address in a revised draft:

1) Clarify the notion of "model misspecification" in the paper. The paper assumes a small set of true parameter-data pairs, whereas classical statistical misspecification does not. Reviewers found this confusing in contrast to Walker (2013) and other references (Wehenkel et al., 2025; Ward et al., 2022). In addition, authors should add the reference to inductive RoPE (Senouf et al., NeurIPS 2025) as having a similar problem formulation with calibration pairs, along with other related literature (see Reviewer Qmdc's updated comments for additional details).

2) General clarity comments, such as clearly defining the setting and target distribution of their algorithm(s), with justification.
Clarifying the finite-sample story: the method still relies heavily on scarce calibration data reused across tightly coupled components, and some of the stronger claims should be softened accordingly.

3) Expand on additional empirical results discussed during the rebuttal, which played a large role in shaping reviewers' favorable opinion during the discussion.

4) Expand on the limitation section in light of reviewers discussion.

With these changes, I am happy to recommend accepting this paper.